# VR-Sampling: Accelerating Flow Generative Model Training with Variance Reduction Sampling

## Abstract

Recent advancements in text-to-image and text-to-video models, such as Stable Diffusion 3 (SD3), Flux and OpenSora, have adopted rectified flow over traditional diffusion models to enhance training and inference efficiency. SD3 notes increased difficulty in learning at intermediate timesteps but does not clarify the underlying cause. In this paper, we theoretically identify the root cause as a higher variance in the loss gradient estimates at these timesteps, which hinders training efficiency. Furthermore, this high-variance region is significantly influenced by the noise schedulers (i.e., how we add noises to clean images) and data (or latent space) dimensions. Building on this theoretical insight, we propose a Variance-Reduction Sampling (VR-sampling) strategy that samples the timesteps in high-variance region more frequently to enhance training efficiency in flow models. VR-sampling constructs sampling distributions based on Monte Carlo estimates of the loss gradient variance, allowing it to easily extend to different noise schedulers and data dimensions. Experiments demonstrate that VR sampling accelerates training by up to 33% on ImageNet 256 and 50% on ImageNet 512 datasets in rectified flow models. Furthermore, VR-sampling could simplify the hyperparameter tuning of logit-normal sampling introduced in SD3. The code is available anonymously in https://github.com/AnonymousProjects/VR_sampling.git.

## 1 Introduction

Diffusion models (Song et al., 2021b; Ho et al., 2020; Dhariwal & Nichol, 2021; Song et al., 2021a) have emerged as powerful generative models, showing remarkable potential in producing high-quality data across various domains, such as image generation (Rombach et al., 2022), video generation (Blattmann et al., 2023), and molecular design (Abramson et al., 2024). While traditional diffusion models typically rely on a diffusion denoising loss for training, the latest state-of-the-art models—such as Stable Diffusion 3 (SD3) (Esser et al., 2024), Flux[1], and OpenSora[2]—have adopted flow matching training loss (Lipman et al., 2022). This approach offers improved training efficiency and a shorter sampling path, making it a preferred choice for modern generative models.

Generative modeling seeks to approximate and sample from a target probability distribution. A prominent approach, diffusion models (Song et al., 2021b; Ho et al., 2020), generate samples by simulating a stochastic differential equation (SDE) that gradually transforms a simple distribution into the data distribution. Their success is largely attributed to a simple regression-based training objective, which bypasses the need to simulate the SDE during training. Additionally, an ODE exists that has the same marginal probability as the SDE, providing an alternative path for model training. Recent work has extended the use of ODEs in generative modeling. For example, continuous normalizing flows (CNFs) (Lipman et al., 2022) leverage flow matching, a method for directly regressing the ODE drift, akin to the training process in diffusion models. Similarly, (Liu et al., 2022) introduced rectified flow, a simplified ODE-based approach for transporting between two observed distributions. Expanding on these advancements, SD3 (Esser et al., 2024) combines both diffusion and flow formulations to train a state-of-the-art, open-source text-to-image model using flow models. SD3 also highlights a key

---

[1] https://github.com/black-forest-labs/flux.git
[2] https://hpcaitech.github.io/Open-Sora/

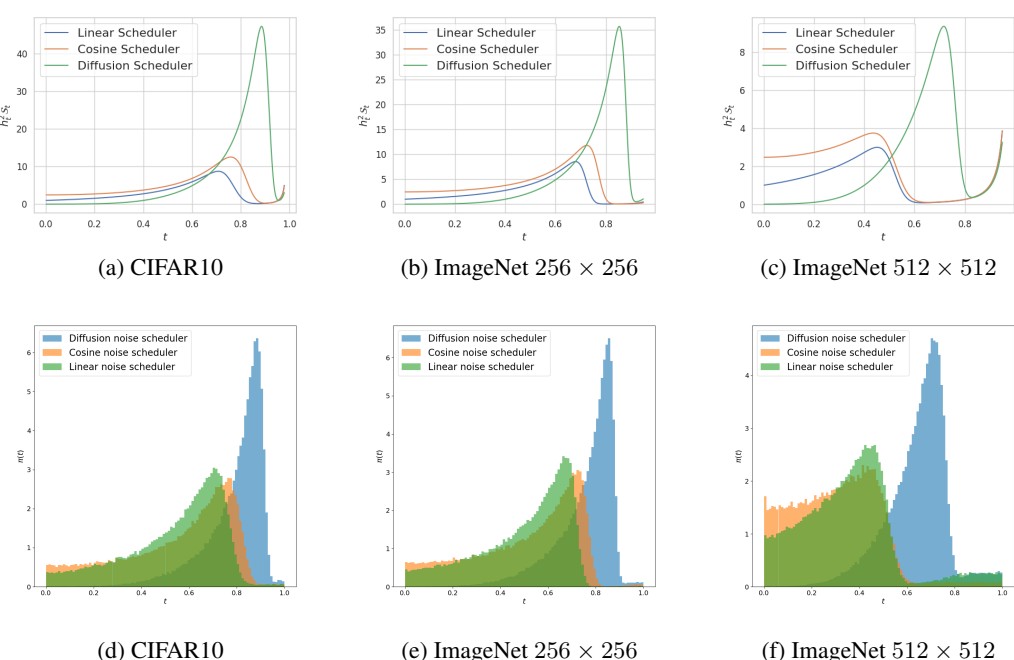

Figure 1: Here $h_t^2 \mathcal{S}_t$ represents the derived upper bound on the variance of conditional loss gradient estimates. Note that $t = 0$ indicates the pure Gaussian noise, and $t = 1$ refers to the data. The curves are generated using Monte Carlo simulations under three noise schedulers: linear (Liu et al., 2022), cosine (Nichol & Dhariwal, 2021), and traditional diffusion (Ho et al., 2020). (a)-(c) display these curves for datasets with different dimensionalities: CIFAR10 with $3 \times 32 \times 32$, ImageNet 256 with latent spaces of $4 \times 32 \times 32$, and ImageNet 512 with latent spaces of $4 \times 64 \times 64$. (d)-(f) presents the probability density function $\pi(t)$ of VR-sampling distributions.

challenge in rectified flow, noting that the target ODE drift is more difficult to learn at intermediate timesteps. To address this, SD3 introduces logit-normal sampling and mode sampling to sample intermediate timesteps more frequently, thus improving the training efficiency of flow models.

In this work, we delve into a deeper understanding of *why the target ODE drift function is particularly challenging to learn at intermediate timesteps*. In the optimization of diffusion models or flow models, techniques such as conditional score matching or conditional flow matching losses are utilized, as we can only obtain the ground truth regression target when conditioned on the given samples. It has been proved in (Lipman et al., 2022) that the conditional and non-conditional losses are identical, up to a constant that is independent of optimization parameters—making the gradients of conditional loss an unbiased estimator of those from non-conditional losses. However, the practical optimization process often involves stochastic estimates of the loss, leading to discrepancies between optimizing conditional loss and unconditional loss. We theoretically analyze the variance of conditional loss gradient estimates and demonstrate that at intermediate timesteps, it exhibits larger variances, and thus influences the training stability of these timesteps. Furthermore, we identify that the high-variance regions closely depend on the *noise schedulers* (which dictate how noise is added to clean images to achieve a pure Gaussian distribution) and the *dimensions of data or latent space*. In Fig. 1 (a)-(c), we present our derived upper bound on the variance of conditional loss gradient estimates under different noise schedulers and datasets. The results show that the high-variance regions vary depending on both the noise scheduler and the dataset dimensionality.

Building on this theoretical analysis, we propose a Variance Reduction-based sampling strategy (VR-sampling) that samples the high-variance regions more frequently to accelerate the training. VR-sampling distributions are constructed using Monte Carlo simulations of the derived upper bounds on the variance and can adjust according to the noise schedulers and data dimensions. The probability density function $\pi(t)$ of simulated VR-sampling distributions are presented in Fig.1 (d)-(f). Unlike previous acceleration methods, such as those in (Hang et al., 2023; Choi et al., 2022; Wang et al., 2024), which primarily focus on noise schedulers and often identify important timesteps heuristically,

VR-sampling provides a more precise identification of high-variance regions depending on various noise schedulers and data dimensions. Experiments show that it significantly accelerates flow model training across different noise schedulers and data dimensions. Futhermore, it can also simplify the hyperparameter choices of logit-normal sampling introduced in SD3 training. In summary, our contributions are as follows:

- We derive an upper bound on the variance of loss gradient estimates and theoretically prove that the convergence rate of the loss value during training depends on this bound. Our analysis reveals that high-variance regions are strongly influenced by the choice of noise schedulers and the dimensionality of the data or latent space.

- Leveraging this insight, we propose a Variance-Reduction Sampling (VR-sampling) strategy that prioritizes sampling from high-variance regions to accelerate training. The VR-sampling distributions are constructed via Monte Carlo simulations of the variance bounds and easily adapt to different noise schedulers and data dimensions.

- Extensive experiments on the ImageNet 256 and 512 datasets show that VR-sampling accelerate training by 33% and 50%, respectively, with linear noise scheduler, by around 35% with cosine noise scheduler, and by up to 50% with diffusion noise scheduler. Furthermore, VR-sampling simplifies hyperparameter tuning for logit-normal sampling as introduced in SD3. Employing logit-normal sampling with our recommended hyperparameters, detailed in Table 3, achieves a 38% speedup over the default choice on ImageNet 512 with linear noise scheduler.

## 2 PRELIMINARIES

In this section, we provide a brief overview of flow-based generative models, discussing their loss functions and noise schedulers. We also show their connection to diffusion models.

**Flow Models**   Denote $x_1$ as a random variable distributed according to some unknown data distribution $q(x_1)$. Let $p_t$ be a probability path generated by a velocity field $u_t$ such that $p_0$ is a simple distribution, e.g., the standard normal distribution $p_0(x) = \mathcal{N}(x|0, I)$, and $p_1 = q$. The velocity field $u_t$ defines a time-dependent flow, which can be described by an ODE: $\mathrm{d}x_t = u_t(x_t, t)\mathrm{d}t$.

A direct way to build the probability path $p_t$ is to regress the velocity field $u_t$ through a neural network $v_\theta$. We can define the corresponding Flow Matching (FM) loss (Lipman et al., 2022) as

$$\mathcal{L}_{FM}(\theta) = \mathbb{E}_{t, p_t(x)}\|v_\theta(x, t) - u_t(x)\|^2, \tag{2.1}$$

where $t \sim \mathcal{U}[0, 1]$ (uniform distribution). Flow matching presents a simple and appealing objective; however, it is impractical to use in practice due to the lack of prior knowledge regarding appropriate choices of $p_t$ and $u_t$.

**Conditional Flow Matching**   A simpler way of constructing a probability path is via a mixture of conditional probability paths. Explicitly, when given a data sample $x_1$, we denote $p_t(x|x_1)$ as a conditional probability path such that it satisfies $p_0(x|x_1) = p_0(x)$ at time $t = 0$ and $p_1(x|x_1)$ at $t = 1$ to be $q(x_1)$. Marginalizing the conditional probability paths over $q(x_1)$ give rise to the marginal probability path $p_t(x) = \int p_t(x|x_1)q(x_1)\mathrm{d}x_1$.

Denote $u_t(x|x_1)$ as a conditional vector field that generates $p_t(x|x_1)$. We also have that by marginalizing over the conditional vector fields in the following sense:

$$u_t(x) = \int u_t(x|x_1)\frac{p_t(x|x_1)q(x_1)}{p_t(x)}\mathrm{d}x_1 = \mathbb{E}_{p(x_1|x)}[u_t(x|x_1)], \tag{2.2}$$

we can obtain the marginal vector field $u_t(x)$. When we choose the conditional probability path as Gaussian, i.e., $p_t(x|x_1) \sim \mathcal{N}(x|a_t x_1, m_t^2 I)$, the velocity field can be solved in closed form. In this case, we could regress the conditional velocity filed $u_t(x|x_1)$ through a neural network $v_\theta$ using the Conditional Flow Matching (CFM) loss:

$$\mathcal{L}_{CFM}(\theta) = \mathbb{E}_{t, q(x_1), p_t(x|x_1)}\|v_\theta(x, t) - u_t(x|x_1)\|^2. \tag{2.3}$$

Besides, it can be theoretically proved that the gradients of $\theta$, $\mathcal{L}_{FM}$ and $\mathcal{L}_{CFM}$ w.r.t. $\theta$ are equal.

Table 1: Different choices of noise scheduler

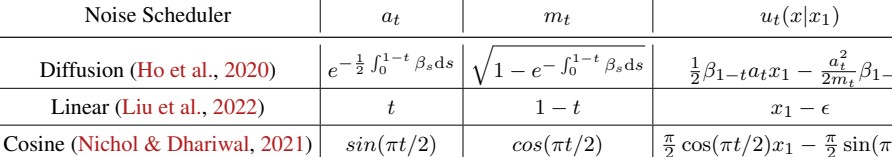

| Noise Scheduler | $a_t$ | $m_t$ | $u_t(x\|x_1)$ |
|---|---|---|---|
| Diffusion (Ho et al., 2020) | $e^{-\frac{1}{2}\int_0^{1-t}\beta_s\mathrm{d}s}$ | $\sqrt{1-e^{-\int_0^{1-t}\beta_s\mathrm{d}s}}$ | $\frac{1}{2}\beta_{1-t}a_tx_1 - \frac{a_t^2}{2m_t}\beta_{1-t}\epsilon$ |
| Linear (Liu et al., 2022) | $t$ | $1-t$ | $x_1 - \epsilon$ |
| Cosine (Nichol & Dhariwal, 2021) | $sin(\pi t/2)$ | $cos(\pi t/2)$ | $\frac{\pi}{2}\cos(\pi t/2)x_1 - \frac{\pi}{2}\sin(\pi t/2)\epsilon$ |

**Theorem 2.1** (Theorem 2 (Lipman et al., 2022)). Assuming that $p_t(x) > 0$ for all $x \in \mathbb{R}^d$ and $t \in [0, 1]$, then up to a constant independent of $\theta$, $\mathcal{L}_{FM}$ and $\mathcal{L}_{CFM}$ are equal. Hence,

$$\nabla_\theta\mathcal{L}_{FM} = \nabla_\theta\mathcal{L}_{CFM}.$$

Now we discuss the construction $u_t(x|x_1)$ for a general family of Gaussian conditional probability paths. Namely, we consider conditional probability paths of the form $p_t(x|x_1) = \mathcal{N}(x|a_tx_1, m_t^2I)$, where $a_0 = 0, m_0 = 1$ and $a_1 = 1, m_1 = 0$. This ensures that all conditional probability paths converge to the same standard Gaussian noise distribution at $t = 0$. The above distribution can also be formulated as

$$x_t = a_tx_1 + m_t\epsilon, \tag{2.4}$$

where $\epsilon \sim \mathcal{N}(0, I)$. This flow provides a vector field that generates the conditional probability path: $\mathrm{d}x_t = u_t(x_t|x_1)\mathrm{d}t$. Then the unique vector field that defines $x_t$ has the following form (Lipman et al., 2022):

$$u_t(x|x_1) = \frac{\dot{m}_t}{m_t}x + \left(\dot{a}_t - a_t\frac{\dot{m}_t}{m_t}\right)x_1. \tag{2.5}$$

Consequently, $u_t(x|x_1)$ generates the Gaussian path $p_t(x|x_1)$.

**Choices of $a_t$ and $m_t$** We could have different choices of $a_t$ and $m_t$. Specifically, when we choose $a_t = t$ and $m_t = 1 - t$, it defines the forward process as straight paths between the data distribution and a standard normal distribution. This is the rectified flow introduced in (Liu et al., 2022) and we call it as **linear noise scheduler** thereafter. Another common choices of $a_t$ and $m_t$ are $sin(\pi t/2)$ and $cos(\pi t/2)$ (Nichol & Dhariwal, 2021) and we call it as **cosine noise scheduler**.

When we choose $a_t = e^{-\frac{1}{2}\int_0^{1-t}\beta_s\mathrm{d}s}$ and $m_t = \sqrt{1-e^{-\int_0^{1-t}\beta_s\mathrm{d}s}}$ where $\beta_t$ could be a linear function between an interval, we have the probability path same as the traditional diffusion models, such as (Ho et al., 2020), and we denote it as **diffusion noise scheduler**. Note that while traditional diffusion models typically use the noise (i.e., $\epsilon$) prediction, flow models instead use velocity (i.e., $u_t$) prediction. Besides, in our experiments, the time definitions are reversed compared to traditional diffusion models: $t = 0$ represents pure Gaussian noise, while $t = 1$ corresponds to the original data points. We list the choices of $a_t$ and $m_t$ and their conditional velocity field $u_t(x|x_1)$ in Table 1 and in this work, we validate our methods under these three noise schedulers.

## 3 METHODS

In Sec. 3.1, we derive an upper bound on the variance of the gradient estimates for conditional flow matching loss at a fixed $x_t$ and $t$. This bound shows how variance is influenced by the noise schedulers and the data dimension (or the latent space dimension in latent flow models). Then in Sec. 3.2, we theoretically prove that the convergence rate of the loss value during training depends on our derived bound. Thus, in Sec. 3.3, we simulate the upper bound of loss gradient variances using the Monte Carlo method and show that the mean variance is more prominent in the intermediate timesteps. Based on these observations, we propose the VR-sampling to mitigate the high variance and thus accelerate training.

### 3.1 GRADIENT VARIANCE DURING TRAINING PROCESS

As demonstrated in Theorem 2.1, the gradients $\nabla_\theta\mathcal{L}_{FM}$ and $\nabla_\theta\mathcal{L}_{CFM}$ are theoretically identical. It is easy to see that given $x$ and $t$, the optimal minimum $v_\theta^*(x, t)$ of Eqn. (2.1) is achieved at $v_\theta^*(x, t) = u_t(x)$ and the optimal minimum of Eqn. (2.3) is achieved when

$$v_\theta^*(x, t) = \mathbb{E}_{q(x_1), p_t(x|x_1)}[u_t(x|x_1)] = \mathbb{E}_{p_t(x_1|x)}[u_t(x|x_1)] = u_t(x).$$

During the gradient descent optimization, by averaging $\nabla_\theta \mathcal{L}_{CFM}(\theta; x_1, x, t)$ over $x_1 \sim q(x_1)$, it becomes a good estimator $\nabla_\theta \mathcal{L}_{FM}(\theta; x, t)$ and will converge to the optimal solution $u_t(x)$. This means that for fixed $x$ and $t$, $\nabla_\theta \mathcal{L}_{CFM}(\theta; x_1, x, t)$ is an unbiased estimator of $\nabla_\theta \mathcal{L}_{FM}(\theta; x, t)$, i.e.,

$$\mathbb{E}_{p_t(x_1|x)}[\nabla_\theta \mathcal{L}_{CFM}(\theta; x_1, x, t)] = \nabla_\theta \mathcal{L}_{FM}(\theta; x, t).$$

However, in practice, flow-based models are trained using stochastic estimates of gradients. For example, in stochastic gradient descents (SGD) optimization, we update the parameters as follows:

$$\theta^{(l)} \leftarrow \theta^{(l-1)} - \eta \nabla_\theta \mathcal{L}_{CFM}(\theta; x_{1,i}, x_{0,i}, t_i),$$

where $x_{1,i} \sim q(x_1), x_{0,i} \sim \mathcal{N}(0, 1), t \sim \text{Uniform}(0, 1)$ and $\eta$ is step size. Thus, in this case, for fixed $t$ and $x$, the variance of $\nabla_\theta \mathcal{L}_{CFM}(\theta; x_1, x, t)$ w.r.t $x_1$, i.e., $\text{Tr}(\text{Cov}_{p(x_1|x)}(\nabla_\theta \mathcal{L}_{CFM}(\theta; x_1, x, t)))$, influences the convergence rate of optimization.

Denote $\mathcal{V}_{x,t} = \text{Tr}(\text{Cov}_{p(x_1|x)}(\nabla_\theta \mathcal{L}_{CFM}(\theta; x_1, x, t)))$. Suppose our target distribution $q$ is normalized, i.e., $\mathbb{E}_{q(x_1)}[x_1] = 0$ and $\frac{1}{d}\mathbb{E}_{q(x_1)}[\|x_1\|^2] = 1$. We could obtain the following lemma:

**Lemma 3.1.** We could upper bound the average total gradient variance as follows:

$$\mathbb{E}_{t,p_t(x)}[\mathcal{V}_{x,t}] \leq 4d\mathbb{E}_{t,p_t(x)}[\|\nabla_\theta v_\theta(x,t)\|^2] \int h_t^2 \left(1 - \mathcal{S}\left(\frac{a_t}{m_t}\right)\right) dt, \tag{3.1}$$

where $h_t = \dot{a}_t - a_t \frac{\dot{m}_t}{m_t}$ and $\mathcal{S}\left(\frac{a_t}{m_t}\right) = \frac{1}{d}\mathbb{E}_{p_t(x)}\|\mathbb{E}_{p_t(x_1|x)}[x_1]\|^2, p_t(x|x_1) \sim \mathcal{N}(a_t x_1, m_t^2 I)$.

The proof of Lemma 3.1 is shown in Appendix C. Here, $\mathcal{S} \in [0, 1]$ is a function related to the signal-to-noise ratio (SNR) $a_t/m_t$, representing the average data separation scale at a given SNR. This concept is also discussed in (Shaul et al., 2023), which highlights the difference between conditional kinetic energy and system kinetic energy. To clarify the notion of data separation, consider a data point $x_{1,i} \sim q(x_1)$. If this point is well-separated at the SNR $a_t/m_t$, then the noised sample $x_t = a_t x_{1,i} + m_t \epsilon$ will be closer to its originating data point $x_{1,i}$ than to any other data point $x_{1,j}$ with $j \neq i$. When all data samples are well-separated, we have $\mathcal{S} = 1$, and it is evident that as $a_t/m_t \to \infty, \mathcal{S} \to 1$.

### 3.2 CONVERGENCE ANALYSIS

Now we have an analysis on the convergence of SGD and show that the convergence rate of the loss value during training depends on $h_t$ and $\mathcal{S}$. Before presenting our theorem, we introduce some necessary assumptions.

**Assumption 3.2.** Suppose $v_\theta(x, t) \in \mathbb{R}^{d_{out}}$ is $L$-smooth and twice differentiable w.r.t. $\theta \in \mathbb{R}^d$ ($d \geq d_{out}$), i.e., there exist a constant $L$ such that

$$\left\|\nabla_\theta^2 (v_\theta(x, t))_i\right\| \leq L, \quad \forall \theta \in \mathbb{R}^d, x, t, i \in d_{out}.$$

For any $\theta \in \mathbb{R}^d$, we further assume $|(v_\theta(x, t) - u_t(x|x_1))_i| \leq \delta < \infty$ is bounded for all $x, x_1$ and $t$.

Assumption 3.2 assumes the smoothness of the neural network, which is quite common, mild, and frequently used in the analysis of general non-convex problems (Guo et al., 2021; Arjevani et al., 2022; Xie et al., 2024). Let $L_\theta$ be a constant that only depends on $\delta, d_{out}$, and $L$. Then under Assumption 3.2, we have $\left\|\nabla_\theta^2 \mathcal{L}_{CFM}(\theta)\right\| \leq L_\theta$ (see its proof in Lemma D.2 in Appendix. D). Finally, we have the following theorem.

**Theorem 3.3** (Linear Convergence). Suppose Assumption 3.2 holds. Consider $\{\theta_k\}_{k \in \mathbb{N}}$ a sequence generated by the SGD algorithm, with a constant stepsize $\eta$ satisfying $0 < \eta < \frac{1}{L_\theta}$. It follows that

$$\mathbb{E}\left[\mathcal{L}_{CFM}(\theta^{k+1})\right] \leq (1 - 2\eta)^k \mathcal{L}_{CFM}(\theta^0) + 2d\left(2\delta + \frac{L}{2}\right)^2 \int h_t^2 \left(1 - \mathcal{S}\left(\frac{a_t}{m_t}\right)\right) dt, \tag{3.2}$$

where $\delta$ and $L$ are the constants that depend on the boundedness and smoothness of $v_\theta(x, t)$.

The proof of Thereom 3.3 is shown in Appendix. D. Although the loss component in the upper bound from Eqn. (3.2) diminishes linearly, a noise term persists within the upper bound. Once the loss reaches a certain threshold, the dominant term in the convergence upper bound effectively becomes noise term. In contrast to conventional optimization theory, the magnitude of this noise is independent of the optimization step size and is solely determined by the sampling policy.

### 3.3 VARIANCE REDUCTION BASED SAMPLING STRATEGY (VR-SAMPLING)

Based on the above theoretical analysis, we design a sampling strategy that aims to reduce the variance of loss gradient estimates. First, since we cannot obtain a closed-form expression for the separation function $\mathcal{S}$, we use the Monte Carlo methods to estimate it. The formulation for the Monte Carlo estimation of $\mathcal{S}$ is

$$\mathcal{S}\left(\frac{a_t}{m_t}\right) = \frac{1}{Md}\sum_{l=1}^{K}\sum_{i=1}^{M}\Big\|\sum_{j=1}^{M}x_{1,j}p_t\left(x_{1,j}|x_{1,i}+\frac{m_t}{a_t}\epsilon_l\right)\Big\|^2, \tag{3.3}$$

where $M$ is the number of data samples and $K$ is the number of Gaussian samples, $x_{1,i} \sim q(x_1)$ and $p_t(x|x_{1,i}) \sim \mathcal{N}(x_{1,i}, m_t^2/a_t^2 I)$. Besides, $p_t(x_{1,i}|x)$ could be calculated based on Bayes' theorem, i.e., $p_t(x_{1,i}|x) = p_t(x|x_{1,i})q(x_{1,i})/p_t(x)$.

We present the simulated values of $\mathcal{S}_t$ and $h_t^2 \mathcal{S}_t$ under different noise schedulers and data dimensions in Fig. 2 and Fig. 1(a)–(c), respectively. In these simulations, the dimensionality of CIFAR10 is $3 \times 32 \times 32$, while the latent space dimensions for ImageNet 256 and ImageNet 512 are $4 \times 32 \times 32$ and $4 \times 64 \times 64$, respectively. We observe in Fig. 2 that the timesteps at which the separation function $\mathcal{S}_t$ transitions from 0 to 1 differ depending on the noise scheduler and data dimension. This indicates that the data samples transition from being completely inseparable to well-separated at different timesteps across various settings. Accordingly, as shown in Fig. 1(a)–(c), the high-variance regions also vary with different noise schedulers and data dimensions.

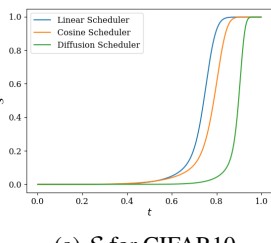 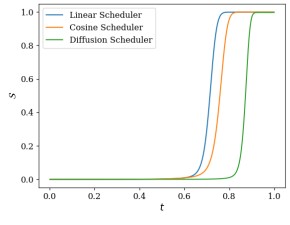 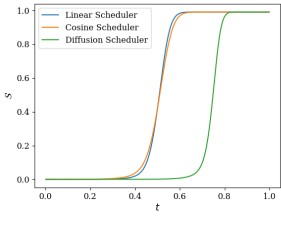

(a) $\mathcal{S}$ for CIFAR10      (b) $\mathcal{S}$ for ImageNet-1k 256      (c) $\mathcal{S}$ for ImageNet-1k 512

Figure 2: Separation Curves $\mathcal{S}$ and $h_t^2 \mathcal{S}_t$ of different noise schedulers under different datasets. Here, $t = 0$ indicates the pure Gaussian noise, and $t = 1$ refers to the data.

In this way, to balance the variance of the loss gradient estimates, we sample timesteps in high-variance region more frequently. Our VR-sampling strategy is designed based on the derived upper bound to create a probability density function (PDF) $\pi(t)$ over $t$. The process involves the following steps: (1) we first normalize the simulated curve by calculating the total area under the curve (via integration) and dividing the curve by the total area; (2) we compute the cumulative sum of the normalized curve to derive the cumulative distribution function (CDF) $F(t)$; (3) Using the CDF, we construct an interpolation function to approximate its inverse $F^{-1}(u), u \sim Uniform(0,1)$; (4) the inverse CDF maps a uniformly distributed random variable to the domain of $t$, ensuring that the resulting samples follow the distribution defined by the probability density function (PDF) $\pi(t)$, where $\pi(t) = F'(t)$. Here the inverse CDF serves as a crucial tool for sampling from the desired density. Besides, as we use Monte Carlo estimation, we cannot reach the exact point where $\mathcal{S}_t = 1$, but as $t \to 1$, $a_t/m_t \to \infty$, causing the simulated variance curve to approach infinity. To address this, we smooth the region near $t = 1$ during normalization to prevent the values from diverging. We present the probability density function $\pi(t)$ of simulated distributions in Fig. 1 (d)-(f).

**Time Consumption on obtaining VR-Sampling** The time required to compute the VR-sampling distributions depends on the number of data samples and Gaussian noise samples selected for the Monte Carlo simulations. For example, on ImageNet $256 \times 256$, we set the number of data samples $M = 200$ and sample 500 Gaussian noise samples to construct $x_t$. This simulation takes approximately 40 minutes using 8 A800 GPUs. This time is significantly shorter and acceptable compared to the several days needed for full training (e.g., for ImageNet 256×256, training for 400K iterations takes approximately 72 hours on 8 A800 GPUs in our setting). More details about the choices of samples under each setting and time consumption are shown in Table 4 in Appendix A.1.

## 4 EXPERIMENTS

We empirically investigate VR-sampling across a range of experimental settings. First, we describe the basic experimental setup in Sec. 4.1, including the datasets, codebases, training details, and evaluation metrics. Next, we present our experimental results in Sec. 4.2 and compare our performance to current state-of-the-art methods for training diffusion models. Finally, we show that our sampling strategy could benefit the hyperparameter choice of logit-normal sampling introduced in SD3 (Esser et al., 2024) in Sec. 4.3.

### 4.1 EXPERIMENT SETTINGS

**Datasets and Codebases** To validate our theoretical analysis, we conduct experiments on datasets of varying dimensions, including CIFAR10, ImageNet-1k 256, and ImageNet-1k 512, and evaluate across three noise schedulers: linear, cosine, and diffusion. For CIFAR10, we use the TorchCFM codebase [3], which leverages an U-Net to train flow models. For ImageNet 256 and ImageNet 512, we employ the SiT codebase [4]. SiT (Ma et al., 2024) is a flow model counterpart to DiT (Peebles & Xie, 2023) and uses the same Transformer architecture. To compare with state-of-the-art methods under diffusion noise scheduler, we test our sampling strategy on the DiT codebase [5].

**Training and Evalutions** To validate the effectiveness of our methods, we adhere to the default training settings across the different codebases and run 200K iterations for CIFAR10 and 400k iterations for ImageNet-1k under each configuration. We use the default training results as the baseline and compare them with our designed sampling strategies. The detailed training settings are provided in Appendix A.2.

For evaluation, we primarily use the Frechet Inception Distance (FID) to assess the performance of the trained models. For CIFAR10, we set the number of sampling steps to 100 and utilize ODE-dopri5 solvers, generating 50K images for FID calculation. For ImageNet 256 and ImageNet 512, we use 250 sampling steps with ODE-dopri5 solvers to generate 10K images for evaluation. We calculate FID under both unconditional generation (with the classifier-free guidance (cfg) scale of 1.0) and conditional generation (with cfg scale of 1.5).

### 4.2 RESULTS

In this section, we show the experimental results across various noise schedulers and data dimensions. Fig. 3 shows the FID curves for the ImageNet-1k 256 dataset under both unconditional (cfg=1.0) and conditional (cfg=1.5) generation when we employ SiT XL/2 network. We observe that, using VR-sampling under linear noise schedule, we achieve the same FIDs (for both conditional and unconditional generation) as the baseline at 400K iterations, but in only 270K iterations—representing a 33% acceleration. Similarly, under a cosine noise schedule, we reach the same FIDs at around 250K iterations, achieving a 38% speedup. Notably, with the diffusion noise schedule, we see an even greater acceleration of 50%, reaching the baseline FIDs at 200K iterations instead of 400K.

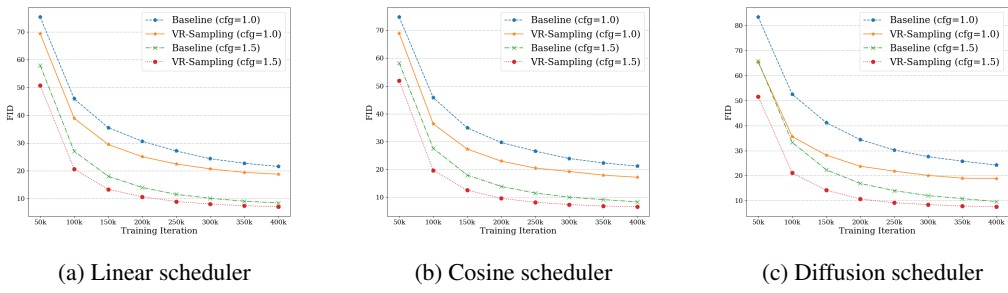

|     (a) Linear scheduler     |     (b) Cosine scheduler     |     (c) Diffusion scheduler     |

Figure 3: FID curves under different noise schedulers for ImageNet-1k 256×256

Fig.4 shows the FID curves for the ImageNet-1k 512 dataset under both unconditional (cfg=1.0) and conditional (cfg=1.5) generation, using the SiT XL/2 network. Remarkably, with the linear

---

[3] https://github.com/atong01/conditional-flow-matching.git

[4] https://github.com/willisma/SiT.git

[5] https://github.com/facebookresearch/DiT.git

noise scheduler, we achieve the same FID as the baseline at 400K iterations in just 200K iterations, delivering a 50% speedup. Similarly, under the cosine noise scheduler, we observe a 33% speedup, and with the diffusion noise scheduler, we once again achieve a 50% speedup. Additionally, we report the FID and Inception Scores (IS) at 400K iterations on the ImageNet-1k dataset in Table 2, further demonstrating the efficiency and performance improvements of our approach.

We present the FID curves for CIFAR10 in Fig. 5 under three noise schedulers. For linear noise schedulers, it achieves the same FIDs as the baseline at 200K iterations in 140K iterations, showing a 30% speedup. It shows 10% speedup under cosine noise schedulers and 46% speedup under diffusion noise schedulers. Note that our methods also generalize to different network size (see Appendix B.1). In addition, our conclusions hold when using fewer sampling steps and other solvers (see the Appendix B.2). We also present some qualitative results in the Appendix B.4.

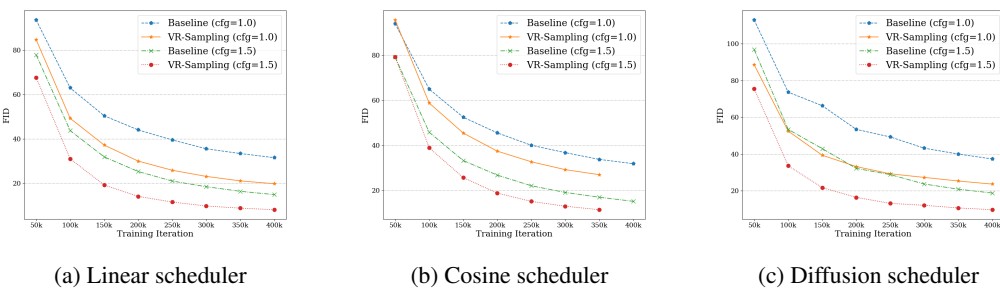

(a) Linear scheduler        (b) Cosine scheduler        (c) Diffusion scheduler

Figure 4: FID curves under different noise schedulers for ImageNet-1k 512×512

Table 2: Comparison of FID and IS metrics for different noise schedulers on ImageNet 256 and ImageNet 512. Here * denotes the results at 350K iterations and other results are at 400K iterations.

| Noise Scheduler | Method | ImageNet 256 | | | | ImageNet 512 | | | |
|---|---|---|---|---|---|---|---|---|---|
| | | FID (↓) | | IS (↑) | | FID (↓) | | IS (↑) | |
| | | cfg=1.0 | cfg=1.5 | cfg=1.0 | cfg=1.5 | cfg=1.0 | cfg=1.5 | cfg=1.0 | cfg=1.5 |
| Linear | Baseline | 21.56 | 8.47 | 71.19 | 157.33 | 31.58 | 14.98 | 49.44 | 103.60 |
| | VR-sampling | **18.79** | **7.10** | **81.93** | **182.01** | **19.89** | **8.19** | **81.92** | **172.59** |
| Cosine | Baseline | 21.18 | 8.34 | 71.87 | 159.40 | 33.74* | 17.0* | 46.25* | 93.00* |
| | VR-sampling | **17.17** | **6.60** | **90.20** | **194.70** | **26.95*** | **11.40*** | **63.14*** | **132.2*** |
| Diffusion | Baseline | 24.27 | 9.69 | 62.78 | 137.88 | 37.29 | 18.71 | 42.54 | 84.13 |
| | VR-sampling | **18.88** | **7.61** | **75.93** | **159.21** | **23.60** | **9.64** | **66.80** | **140.68** |

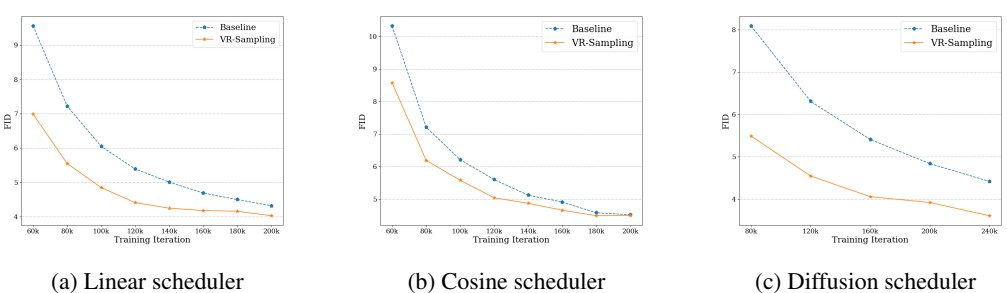

(a) Linear scheduler        (b) Cosine scheduler        (c) Diffusion scheduler

Figure 5: FID curves under different noise schedulers for CIFAR10 datasets

## 4.3 LOGIT-NORMAL SAMPLING

Stable Diffusion 3 (Esser et al., 2024) introduces logit-normal sampling to accelerate training under linear noise scheduler, a method that has been adopted in subsequent training frameworks, such as OpenSora [6] and Flux. The density function for logit-normal sampling is given by:

$$\pi_{ln}(t; m, s) = \frac{1}{s\sqrt{2\pi}} \frac{1}{t(1-t)} \exp\left(-\frac{(\text{logit}(t) - m)^2}{2s^2}\right),$$

---

[6] https://github.com/hpcaitech/Open-Sora.git

where $\text{logit}(t) = \log \frac{t}{1-t}$, with a location parameter $m$ and a scale parameter $s$. In practical training, the values of $m$ and $s$ are usually set to 0 and 1, respectively.

Inspired by our VR-sampling strategy, we explore its potential to inform hyperparameter choices for logit-normal sampling. We set the scale parameter $s$ to 1 and align the location parameter $m$ with the distribution of our VR-sampling. Specifically, we perform a Kolmogorov-Smirnov (KS) test to compare the VR-sampling distribution with various logit-normal distributions. We choose $m$ in steps of 0.25 (that is, $\ldots, -0.75, -0.5, \ldots, 0.5, 0.75, \ldots$), selecting the value of $m$ that minimizes the KS score, thus achieving a closer match between the two distributions. Based on our simulated VR-sampling distributions shown in Fig. 1 (d)-(f), we conclude the optimal choice of $m$ in Table 3 across various noise schedulers and data dimensions. Here we present a range of $m$ as the two distributions cannot be totally matched.

Table 3: Recommended choices of location parameters $m$ under different settings. In our experiments, we set $t = 0$ to be Gaussian noise and $t = 1$ be data samples. In the reverse case where $t = 1$ refers to Gaussian noise and $t = 0$ refers to data, just set $m' = -m$, where $m$ is our listed choices.

| Data dimensions | $3 \times 32 \times 32$ | $4 \times 32 \times 32$ | $4 \times 64 \times 64$ |
|---|---|---|---|
| Linear Scheduler | $0.25 \sim 0.5$ | $0 \sim 0.25$ | $-0.75 \sim -1$ |
| Cosine Scheduler | $0.25 \sim 0.5$ | $0 \sim 0.25$ | $-0.75 \sim -1$ |
| Diffusion Scheduler | $1.25 \sim 1.5$ | $0.75 \sim 1$ | $0.25 \sim 0.5$ |

Experiments demonstrate that logit-normal sampling with our recommended choice of $m$ outlined in Table. 3 could accelerate training much more effectively than default choice of $m = 0$ under linear noise scheduler. Besides, we also find that it is also effective under cosine and diffusion noise schedulers. For example, in ImageNet-1k 512×512 dataset, we determine through Kolmogorov-Smirnov (KS) score comparisons that the optimal choice of $m$ is $-0.75$ under linear noise scheduler. We compare the FID curves under VR-sampling, logit-normal sampling with $m = -0.75$ and $m = 0$ in Fig. 6. The results show that logit-normal sampling with $m = -0.75$ achieves a 47% training speedup over the baseline and a 38% speedup over the default $m = 0$. Moreover, logit normal sampling with $m = -0.75$ achieves a speedup comparable to that of our VR sampling, indicating that it offers an alternative. For future ease of use, if the data dimensions match those specified in Table 3, the logit-normal sampling with the recommended $m$ can be employed directly, eliminating the need to simulate the VR-sampling distribution again.

Another example involves the ImageNet-1k 256×256 dataset with diffusion noise scheduler. We compared the FID curves for VR-sampling, logit-normal sampling with $m' = -0.75$ (in DiT, $m' = -m$), and the current SoTA method—SpeeD (Wang et al., 2024), designed specifically for diffusion training. The SpeeD method was implemented using its official codebase[7]. As illustrated in Fig.7, both VR-sampling and logit-normal sampling with our recommended $m$ value converge slightly faster than SpeeD. Unlike SpeeD, which relies on a complex heuristic sampling and weighting strategy based solely on changes in SNR (as discussed in Theorem 1 of SpeeD), our methods target the fundamental factors affecting training. This makes our methods easily applicable across various noise schedulers and datasets and obtain good performance. In fact, it remains uncertain whether the SpeeD method is still effective when applied to flow model training. We presents additional results for other noise schedulers and data dimensions in Appendix B.3.

## 5  RELATED WORK

**Flow and Diffusion Generative Models**   Diffusion models have emerged as a powerful class of generative models, particularly effective in generating high-quality images. These models operate by defining a forward process, where noise is incrementally added to data, and a reverse process, which denoises the data to generate samples. Building on the principles of diffusion models, Lipman et al. (2022) introduced flow matching, which constructs a probability flow ODE directly to generate samples from learned distributions. Liu et al. (2022) propose rectified flow to learn the transformation between Gaussian noise and a target data distribution, by creating a straight-line path between them. Huang et al. (2024) proposes incorporating correlated noise instead of pure Gaussian noise into deterministic diffusion models, which improves image quality. Similarly, Heitz et al. (2023) introduces iterative $\alpha$-blending, which blends and deblends samples between two densities. Further

---

[7]https://github.com/kaiwang960112/SpeeD.git

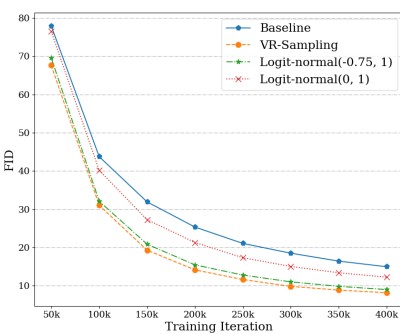
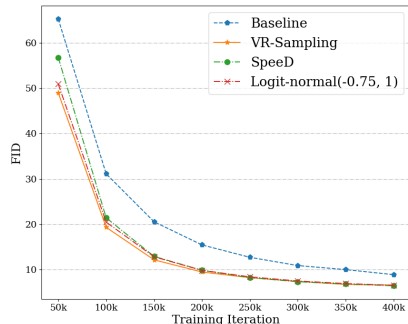

Figure 6: FIDs (cfg=1) with linear scheduler in SiT XL/2 models for ImageNet 512.

Figure 7: FIDs (cfg=1) with diffusion scheduler in DiT XL/2 models for ImageNet 256.

advancing this line of work, SD3 (Esser et al., 2024) scales rectified flow to large-scale text-to-image models, pushing the boundaries of diffusion-based generative models.

**Training Acceleration in Diffusion Models**  To accelerate the training of diffusion models, P2 (Choi et al., 2022) and Min-SNR (Hang et al., 2023) introduce two re-weighting methods. P2 weighting prioritizes learning from key noise levels determined by observations, while Min-SNR adjusts timestep loss weights based on clamped signal-to-noise ratios to balance conflicts. There are also two re-sampling methods: Log-Normal (Karras et al., 2022) and CLTS (Xu et al., 2024) and a method combining re-weighting and re-sampling: SpeeD (Wang et al., 2024). Log-Normal (Karras et al., 2022) assigns high sampling probabilities at intermediate timesteps. CLTS (Xu et al., 2024) proposes a curriculum learning based timestep schedule, which leverages the noise rate as an explicit indicator of the learning difficulty and gradually reduces the training frequency of easier timesteps, thus improving the training efficiency. SpeeD (Wang et al., 2024) design an asymmetric sampling strategy that reduces the frequency of steps from the convergence area while increasing the sampling probability for steps from other areas. Additionally, they propose a weighting strategy to emphasize the importance of time steps with rapid-change process increments. All above methods only consider the impact of noise schedulers on training and most strategies only rely on the change of SNR.

Our work builds on the concept of importance sampling to reduce gradient variance. As demonstrated by (Katharopoulos & Fleuret, 2018), importance sampling can focus computational efforts on informative examples, effectively reduce gradient variance during training, and accelerate convergence. Additionally, Jeha et al. explores variance reduction through a $k$-th order Taylor expansion applied to the diffusion training objective and its gradient. In contrast to (Jeha et al.), our approach directly identifies and targets high-variance regions during training, sampling them more frequently to reduce variance and improve efficiency, without relying on approximations as in (Jeha et al.).

# 6 CONCLUSION

In this work, we propose VR-sampling to accelerate flow generative model training by sampling timesteps in the high-variance region more frequently. We begin by theoretically analyzing the upper bound on the variance of loss gradient estimates and identify that the high-variance region is closely correlated with the noise schedulers and data (or latent space) dimensions. Based on this theoretical insight, we design VR-sampling distributions based on Monte Carlo simulations of our derived variance upper bounds. We validate the efficiency of VR-sampling under different noise schedulers and different datasets. Furthermore, VR-sampling could simplify hyperparameter tuning for logit-normal sampling proposed in SD3 and experiments show that logit-normal sampling with our recommended choice of $m$ also has a good training speedup.

**Limitations and future work**  Due to the limitation of GPU resources, we do not test our sampling strategy on larger model size, such as stable diffusion 3 and video generation models. Extending our approach to larger-scale models, where computational demands are significantly higher, would greatly enhance the value of our methods. Additionally, we observe that our theoretical analysis can be applied to consistency flow model training, allowing for the design of a sampling strategy to accelerate consistency training. We will leave this exploration for future work.

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

## A TRAINING DETAILS

### A.1 DETAILS ON MONTE CARLO SIMULATIONS

Denote the number of samples as $M$ and the number of Gaussian samples are $K$. Besides, we divide the time range $[0, 1]$ into 1000 discrete timesteps. We implement the Monte Carlo simulation on GPUs, as most of the calculations involve matrix operations. The time consumption (in seconds) to compute $\mathcal{S}_t$ for each timestep shown in Table. 4. We can find that the computation time are highly dependent on the choice of $M$ and $K$. However, since the computations for each timestep are independent, they can be executed in parallel across multiple GPUs to reduce the time consumption. Furthermore, as shown in Fig. 8, increasing $M$ and $K$ makes the high-variance region more pronounced, but also increases time consumption. To balance this trade-off, we typically set $M = 200$ and $K = 500$ which has been experimentally shown to work well.

Table 4: Time consumption with different $M$ and $K$

| $M$ | $K$ | sec/per timestep |
|-----|-----|------------------|
| 100 | 200 | $\approx 3.8$s |
| 200 | 500 | $\approx 17.5$s |
| 500 | 1000 | $\approx 101.5$s |

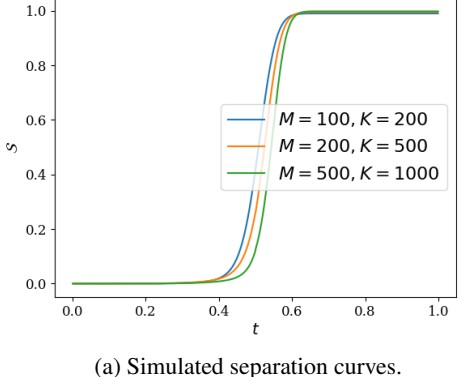

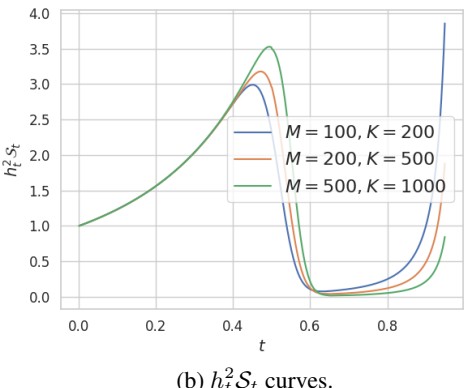

(a) Simulated separation curves.  (b) $h_t^2 \mathcal{S}_t$ curves.

Figure 8: Separation curves and $h_t^2 \mathcal{S}_t$ under different $M$ and $K$ with linear noise scheduler in ImageNet 512×512.

### A.2 TRAINING SETTINGS

In this part, we present the detailed setting of training across different settings. For experiments related with CIFAR10, we implement the cosine noise scheduler and diffusion noise scheduler in the TorchCFM codebase [8] and choose the "icfm" model as the model under linear noise scheduler. We run each setting using one A800 GPU and the explicit training details are as follows:

Table 5: Training Parameters for CIFAR10

| Parameter | Value |
|-----------|-------|
| Learning Rate (`--lr`) | 2e-4 |
| EMA Decay (`--ema_decay`) | 0.9999 |
| Batch Size (`--batch_size`) | 128 |
| Total Steps (`--total_steps`) | 200001 |

---

[8] https://github.com/atong01/conditional-flow-matching.git

For training in SiT [9] and DiT [10] codebase, we run each setting using A800 GPUs and the explicit training details are:

Table 6: Training parameters for ImageNet 256 and 512

|  | ImageNet 256 | ImageNet 256 | ImageNet 512 |
|---|---|---|---|
| Model | SiT-S/2 | SiT-XL/2, DiT-XL/2 | SiT-XL/2 |
| Number of GPUs | 4 | 8 | 8 |
| Training Iterations | 700K | 400K | 400K |
| Image size(`--image-size`) | 256 | 256 | 512 |
| Number of classes (`--num-classes`) | 1000 | 1000 | 1000 |
| Global batch size(`--global-batch-size`) | 256 | 256 | 128 |
| VAE (`--vae`) | ema | ema | ema |
| CFG scale (`--cfg-scale`) | 4.0 | 4.0 | 4.0 |
| Optimizer | AdamW | AdamW | AdamW |
| Learning rate | 1e-4 | 1e-4 | 1e-4 |
| Weight decay | 0 | 0 | 0 |

## B MORE EXPERIMENTAL RESULTS

### B.1 GENERALIZE TO DIFFERENT NETWORK SIZE

The results presented in Sec. 4.2 show the performance across various data dimensions and noise schedules using SiT XL/2, which contains 675 million parameters. In this part, we demonstrate that our sampling strategy is equally effective with SiT S/2, which has 33 million parameters. We test this case under ImageNet-1k 256 datasets and report the FIDs of unconditional generation. We could also observe around 35% speedups under the linear and cosine noise scheduler and 50% speedup under diffusion noise scheduler.

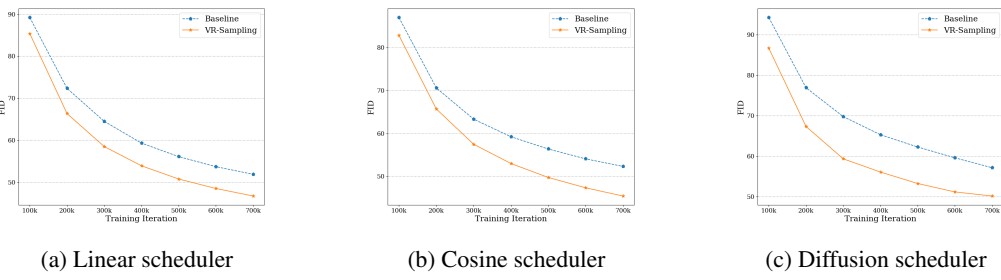

| (a) Linear scheduler | (b) Cosine scheduler | (c) Diffusion scheduler |
|---|---|---|

Figure 9: FID curves (cfg=1.0) under different noise schedulers when using SiT S/2

### B.2 RESULTS ON FEWER SAMPLING STEPS

In Sec. 4.2, we calculated the FID values using 250 sampling steps with the ODE-dopri5 solvers. In this section, we demonstrate that our conclusions remain valid even with a reduced number of sampling steps using alternative solvers. We reduced the sampling steps to 30 and employed the ODE-Euler solvers. The results shown in Table. 7 confirm that with fewer sampling steps and different solvers, we can achieve comparable, and sometimes superior, improvements in both FID and IS metrics.

### B.3 CHOICES OF $m$

In this part, we provide additional experimental results about the choices of $m$. As it is time-consuming to validate every setting, we only choose part of settings and try to cover all noise

---

[9] https://github.com/willisma/SiT.git
[10] https://github.com/facebookresearch/DiT.git

Table 7: Comparison of FID and IS metrics under different sampling step and method on ImageNet 256. Here we present the results when cfg=1.0 at 400K iterations.

| Noise Scheduler | Method | ImageNet 256 | | | |
|---|---|---|---|---|---|
| | | FID (↓) | | IS (↑) | |
| | | ODE-dopri5 250 steps | ODE-Euler 30 steps | ODE-dopri5 250 steps | ODE-Euler 30 steps |
| Linear | Baseline | 21.56 | 24.38 | 71.19 | 68.74 |
| | VR-sampling | 18.79 | 21.39 | 81.93 | 80.42 |
| | **improve** | **2.77** | **2.99** | **10.74** | **11.68** |
| Cosine | Baseline | 21.18 | 26.00 | 71.87 | 65.22 |
| | VR-sampling | 17.17 | 21.55 | 90.20 | 81.95 |
| | **improve** | **4.01** | **4.45** | **18.33** | **16.73** |
| Diffusion | Baseline | 24.27 | 33.47 | 62.78 | 51.95 |
| | VR-sampling | 18.88 | 26.85 | 75.93 | 62.69 |
| | **improve** | **5.39** | **6.62** | **13.15** | **10.74** |

schedulers and datasets. In ImageNet 512×512, we choose linear noise scheduler and present the FID (cfg = 1) curves under SiT XL/2 networks in Fig. 10 (a). In this case, the KS-scores between our simulated curves and logit-normal sampling is shown in Fig. 10 (b). We can find that in this case, the recommended choices of $m$ is $-1 \sim -0.75$ outlined in Table. 3 has smaller KS scores and the FID curves converge faster than other choices. Specifically, logit-normal sampling with our recommended choices achieves a 47% training speedup over the baseline and a 38% speedup over the default m = 0.

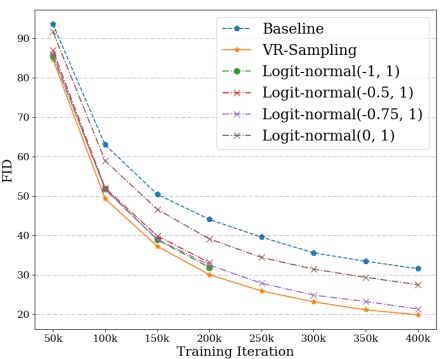
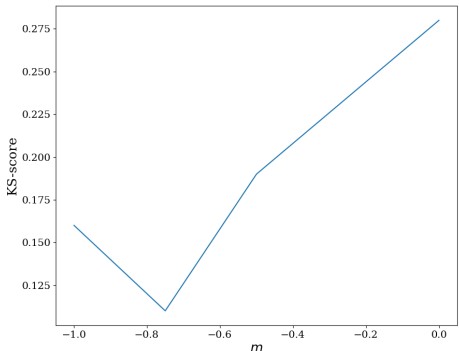

(a) FID curves under linear noise scheduler.

(b) KS scores under different $m$.

Figure 10: FID curves and KS score under linear noise scheduler in ImageNet 512×512.

In ImageNet $256 \times 256$, we choose the cosine noise scheduler and present the FID (cfg = 1) curves under SiT XL/2 networks in Fig. 11 (a). In this case, the KS-scores between our simulated curves and logit-normal sampling is shown in Fig. 11 (b). We can find that in this case, the recommended choices of $m$ is $0 \sim 0.25$ outlined in Table. 3 has smaller KS scores and the FID curves converge faster than other choices. Specifically, logit-normal sampling with our recommended choices of $m$ achieves a 37.5% training speedup over the baseline and an about 25% training speedup over the other choices of $m$, such as $m = -0.5$ or $m = 0.5$.

In CIFAR10, we choose the diffusion noise scheduler and present the FID curves under Unet in Fig. 12 (a). In this case, the KS-scores between our simulated curves and logit-normal sampling is shown in Fig. 12 (b). We can find that in this case, the recommended choices of $m$ is $1.25 \sim 1.5$ outlined in Table. 3 has smaller KS scores and the FID curves converge faster than other choices.

### B.4    QUALITATIVE RESULTS

In this part, we present some qualitative results in Fig. 13, Fig. 14 and Fig. 15.

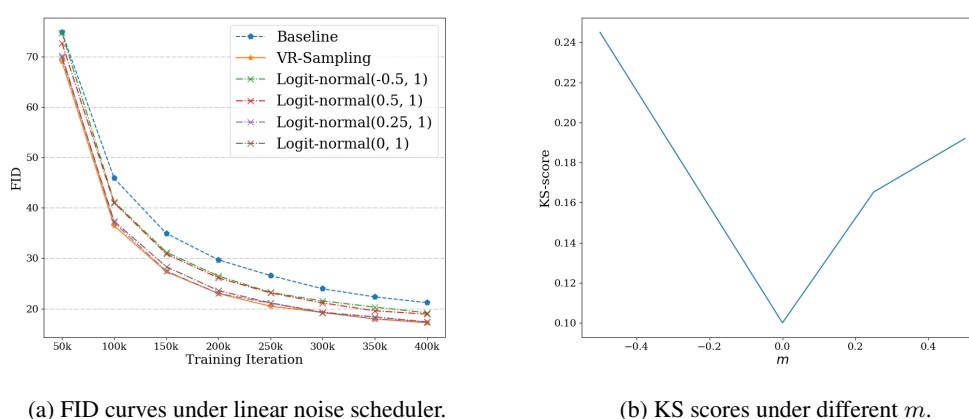

(a) FID curves under linear noise scheduler.

(b) KS scores under different $m$.

Figure 11: FID curves and KS score under cosine noise scheduler in ImageNet $256\times256$.

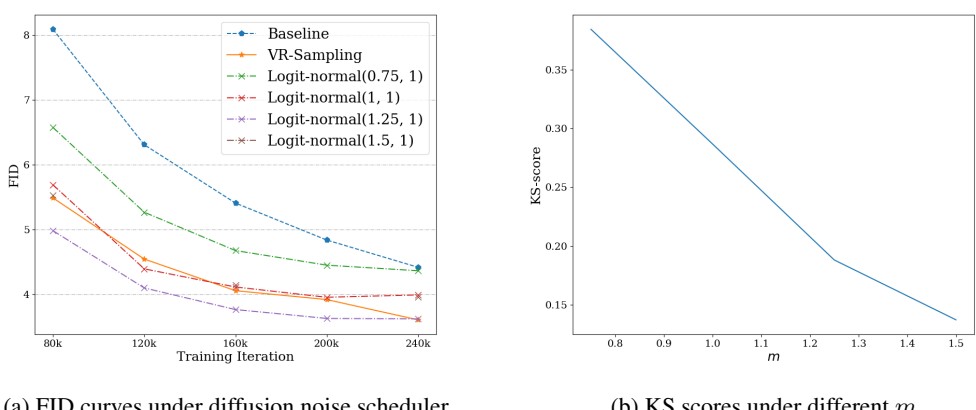

(a) FID curves under diffusion noise scheduler.

(b) KS scores under different $m$.

Figure 12: FID curves and KS score under diffusion noise scheduler in CIFAR10.

## B.5 MORE QUANTITATIVE RESULTS

We include additional experimental results showing FID values with more training iterations (900K iterations) in Fig. 19. We also show the gradient curve in Fig. 20. Additionally, we also show the results with higher cfg values in Fig. 21. In this figure, we present the FID and IS curves under cfg=3.0 and cfg=4.5 under linear noise schedulers for ImageNet256 datasets. A high classifier-free guidance (cfg) scale introduces a trade-off between fidelity (alignment to conditional input) and diversity/realism. Specifically:

- Inception Score (IS) improves with higher cfg values as the generated images become more distinct and coherent with their class features, resulting in higher confidence predictions.

- FID, however, often suffers at high cfg values because the generated images deviate from the natural statistics of the dataset due to overfitting to the class guidance.

In Fig. 21 (a), for higher cfg values (e.g., cfg=3.0 and cfg=4.5), the trends of the FID curves differ from those at lower cfg values (e.g., cfg=1.0 and cfg=1.5). Initially, our method outperforms the baseline in terms of FID, but with longer training, the FID eventually becomes worse than the baseline. This occurs because the model fits the data better but becomes overly aligned to the class conditions at higher cfg values, reducing diversity and realism.

However, as shown in Fig. 21 (b), the Inception Scores (IS) consistently outperform the baseline methods across all cfg values, indicating better semantic alignment and class fidelity. Additionally, at higher cfg scales, the differences between our method and the baseline are more pronounced,

highlighting the advantages of VR-sampling in these settings. We also present the results on LSUN bedroom-256 datasets in Table. 8.

Table 8: Comparison of Baseline and VR-sampling across iterations under LSUN Bedroom 256 datasets.

| Iterations | Baseline | VR-sampling |
|---|---|---|
| 50k | 11.94 | 10.05 |
| 100k | 4.91 | 4.50 |
| 150k | 4.35 | 4.03 |
| 200k | 4.23 | 4.01 |
| 250k | 4.10 | 3.91 |
| 300k | 4.09 | 3.98 |
| 350k | 4.16 | 4.04 |

## C  PROOF OF LEMMA 3.1

We begin to prove Lemma 3.1. We have that

$$\text{Cov}_{p(x_1|x)}\left(\nabla_\theta\|v_\theta(x,t) - u_t(x|x_1)\|^2\right)$$

$$= \text{Cov}_{p(x_1|x)}\left(\nabla_\theta\|v_\theta(x,t)\|^2 - (\nabla_\theta v_\theta(x,t))^T u_t(x|x_1)\right)$$

$$= \mathbb{E}_{p(x_1|x)}\left[\left(\nabla_\theta\|v_\theta(x,t) - u_t(x|x_1)\|^2\right)\left(\nabla_\theta\|v_\theta(x,t) - u_t(x|x_1)\|^2\right)^T\right]$$

$$\quad - \mathbb{E}_{p(x_1|x)}\left[\left(\nabla_\theta\|v_\theta(x,t) - u_t(x|x_1)\|^2\right)\right]\mathbb{E}_{p(x_1|x)}\left[\left(\nabla_\theta\|v_\theta(x,t) - u_t(x|x_1)\|^2\right)^T\right].$$

For the second term in the RHS, we have

$$\mathbb{E}_{p(x_1|x)}\left[\left(\nabla_\theta\|v_\theta(x,t) - u_t(x|x_1)\|^2\right)\right]$$

$$= \mathbb{E}_{p(x_1|x)}\left[\left(\nabla_\theta\|v_\theta(x,t) - u_t(x|x_1)\|^2\right)\right]$$

$$= \nabla_\theta\|v_\theta(x,t)\|^2 - 2(\nabla_\theta v_\theta(x,t))^T\mathbb{E}_{p(x_1|x)}[u_t(x|x_1)]$$

$$\overset{(a)}{=} \nabla_\theta\|v_\theta(x,t)\|^2 - 2(\nabla_\theta v_\theta(x,t))^T u_t(x)$$

$$= \nabla_\theta\|v_\theta(x,t) - u_t(x)\|^2,$$

where (a) is based on Eqn. (2.2). Then, we have

$$\text{Cov}_{p(x_1|x)}\left(\nabla_\theta\|v_\theta(x,t) - u_t(x|x_1)\|^2\right)$$

$$= (\nabla_\theta\|v_\theta(x,t)\|^2)(\nabla_\theta\|v_\theta(x,t)\|^2)^T - 4(\nabla_\theta\|v_\theta(x,t)\|^2)\mathbb{E}_{p(x_1|x)}[u_t(x|x_1)]^T\nabla_\theta v_\theta(x,t)$$

$$\quad + 4(\nabla_\theta v_\theta(x,t))^T\mathbb{E}_{p(x_1|x)}[u_t(x|x_1)u_t(x|x_1)^T](\nabla_\theta v_\theta(x,t))$$

$$\quad - (\nabla_\theta\|v_\theta(x,t)\|^2)(\nabla_\theta\|v_\theta(x,t)\|^2)^T + 4(\nabla_\theta\|v_\theta(x,t)\|^2)u_t(x)^T\nabla_\theta v_\theta(x,t)$$

$$\quad - 4(\nabla_\theta v_\theta(x,t))^T u_t(x)u_t(x)^T(\nabla_\theta v_\theta(x,t))$$

$$= 4(\nabla_\theta v_\theta(x,t))^T\mathbb{E}_{p(x_1|x)}[u_t(x|x_1)u_t(x|x_1)^T - u_t(x)u_t(x)^T](\nabla_\theta v_\theta(x,t)).$$

In this way,

$$\mathcal{V}_{x,t} = \text{Tr}(\text{Cov}_{p(x_1|x)}(\nabla_\theta\mathcal{L}_{CFM}(\theta; x_1, x, t)))$$

$$= \text{Tr}\left(4(\nabla_\theta v_\theta(x,t))^T\mathbb{E}_{p(x_1|x)}[u_t(x|x_1)u_t(x|x_1)^T - u_t(x)u_t(x)^T](\nabla_\theta v_\theta(x,t))\right)$$

$$= 4\text{Tr}\left(\mathbb{E}_{p(x_1|x)}[u_t(x|x_1)u_t(x|x_1)^T - u_t(x)u_t(x)^T](\nabla_\theta v_\theta(x,t))(\nabla_\theta v_\theta(x,t))^T\right)$$

$$= 4\left\langle\mathbb{E}_{p(x_1|x)}[u_t(x|x_1)u_t(x|x_1)^T - u_t(x)u_t(x)^T], (\nabla_\theta v_\theta(x,t))(\nabla_\theta v_\theta(x,t))^T\right\rangle_F$$

$$\leq 4\|\mathbb{E}_{p(x_1|x)}[u_t(x|x_1)u_t(x|x_1)^T - u_t(x)u_t(x)^T]\|_F\|(\nabla_\theta v_\theta(x,t))(\nabla_\theta v_\theta(x,t))^T\|_F$$

$$\leq 4\mathbb{E}_{p(x_1|x)}[\|u_t(x|x_1)u_t(x|x_1)^T - u_t(x)u_t(x)^T\|_F]\|(\nabla_\theta v_\theta(x,t))(\nabla_\theta v_\theta(x,t))^T\|_F$$

$$= 4\|\nabla_\theta v_\theta(x,t)\|^2\mathbb{E}_{p(x_1|x)}[\|u_t(x|x_1)\|^2 - \|u_t(x)\|^2].$$

Recall that we have

$$u_t(x|x_1) = \frac{\dot{m}_t}{m_t}x + \left(\dot{a}_t - a_t\frac{\dot{m}_t}{m_t}\right)x_1.$$

Denote $g_t = \frac{\dot{m}_t}{m_t}$ and $h_t = \left(\dot{a}_t - a_t\frac{\dot{m}_t}{m_t}\right)$. Then we have that

$$
\begin{aligned}
\mathbb{E}_{t,q(x_1),p_t(x|x_1)}[\|u_t(x|x_1)\|^2] &= \mathbb{E}_{t,q(x_1),p_t(x|x_1)}[\|g_t x + h_t x_1\|^2] \\
&\stackrel{(a)}{=} \mathbb{E}_{t,q(x_1),p_0(x_0)}[\|g_t m_t x_0 + g_t a_t x_1 + h_t x_1\|^2] \\
&= \mathbb{E}_{t,q(x_1),p_0(x_0)}\left[\left\|\dot{m}_t x_0 + \dot{m}_t a_t/m_t x_1 + \left(\dot{a}_t - a_t\frac{\dot{m}_t}{m_t}\right)x_1\right\|^2\right] \\
&= \mathbb{E}_{t,q(x_1),p_0(x_0)}[\|\dot{m}_t x_0 + \dot{a}_t x_1\|^2] \\
&\stackrel{(b)}{=} d \cdot \mathbb{E}_t[\dot{m}_t^2 + \dot{a}_t^2]
\end{aligned}
$$

where $(a)$ is because $x = m_t x_0 + a_t x_1$ and $x_0$ and $x_1$ is independent, (b) is because $x_0 \sim \mathcal{N}(0, I)$ and we assume $q$ is normalized. We also have that

$$
\begin{aligned}
\mathbb{E}_{t,p_t(x)}[\|u_t(x)\|^2] &= \mathbb{E}_{t,p_t(x)}\left[\|\mathbb{E}_{p_t(x_1|x)}u_t(x|x_1)\|^2\right] \\
&= \mathbb{E}_{t,p_t(x)}\left[\|g_t x + h_t \mathbb{E}_{p_t(x_1|x)}[x_1]\|^2\right] \\
&= \mathbb{E}_{t,p_t(x)}\left[g_t^2\|x\|^2 + 2g_t h_t \mathbb{E}_{p_t(x_1|x)}[x_1] + h_t^2\|\mathbb{E}_{p_t(x_1|x)}[x_1]\|^2\right] \\
&= \mathbb{E}_{t,p_t(x)}\left[g_t^2\|x\|^2 + 2g_t h_t \mathbb{E}_{p_t(x_1|x)}[x_1] + h_t^2 d - h_t^2 d + h_t^2\|\mathbb{E}_{p_t(x_1|x)}[x_1]\|^2\right] \\
&= \mathbb{E}_{t,p_t(x)}\left[g_t^2\|x\|^2 + 2g_t h_t \mathbb{E}_{p_t(x_1|x)}[x_1] + h_t^2 d\right] - \mathbb{E}_{t,p_t(x)}[h_t^2 d - h_t^2\|\mathbb{E}_{p_t(x_1|x)}[x_1]\|^2].
\end{aligned}
$$

For the first term in RHS, we have

$$
\begin{aligned}
\mathbb{E}_{t,p_t(x)}\left[g_t^2\|x\|^2 + 2g_t h_t \mathbb{E}_{p_t(x_1|x)}[x_1] + h_t^2 d\right] &= \mathbb{E}_{t,p_t(x),p_t(x_1|x)}\left[g_t^2\|x\|^2 + 2g_t h_t x_1 + h_t^2 d\right] \\
&= \mathbb{E}_{t,p_t(x),p_t(x_1|x)}\left[g_t^2\|x\|^2 + 2g_t h_t x_1 + h_t^2\|x_1\|^2\right] \\
&= \mathbb{E}_{t,p_t(x),p_t(x_1|x)}[\|g_t x + h_t x_1\|^2] \\
&= \mathbb{E}_{t,q(x_1),p_t(x|x_1)}[\|u_t(x|x_1)\|^2].
\end{aligned}
$$

Denote $\mathcal{S}\left(\frac{a_t}{m_t}\right) = \frac{1}{d}\mathbb{E}_{p_t(x)}\|\mathbb{E}_{p_t(x_1|x)}[x_1]\|^2$. Thus, the second term in RHS is

$$\mathbb{E}_{t,p_t(x)}[h_t^2 d - h_t^2\|\mathbb{E}_{p_t(x_1|x)}[x_1]\|^2] = \mathbb{E}_t\left[dh_t^2\left(1 - \mathcal{S}\left(\frac{a_t}{m_t}\right)\right)\right].$$

Combining the above results, we have

$$\mathbb{E}_{t,p(x),p(x_1|x)}[\|u_t(x|x_1)\|^2 - \|u_t(x)\|^2] = \mathbb{E}_t\left[dh_t^2\left(1 - \mathcal{S}\left(\frac{a_t}{m_t}\right)\right)\right],$$

which completes the proof of Lemma 3.1.

# D  CONVERGENCE GUARANTEE

To prove the main results, we first provide a auxiliary lemma.

**Lemma D.1** (Polyak-Łojasiewicz (PL) Inequality)**.** Suppose $v_\theta(x, t) \in \mathbb{R}^{d_{out}}$ is differentiable w.r.t. $\theta \in \mathbb{R}^d$ and $d \geq d_{out}$. For all $\theta \in \mathbb{R}^d$, we have

$$\mathcal{L}_{CFM}(\theta) \leq \frac{1}{4}\left(\|\nabla_\theta \mathcal{L}_{CFM}(\theta)\|^2 + \text{Var}\left(\nabla_\theta \mathcal{L}_{CFM}(\theta; x_1, x, t)\right)\right).$$

*Proof.* We separate the last layer learnable bias term from $\theta$, namely, $\theta := [\theta_{-b}, b]$, where $b$ is the learnable bias of the last layer. And hence, there exists $v_{\theta_{-b}}(x, t)$ such that $v_\theta(x, t) = v_{\theta_{-b}}(x, t) + b$, we have

$$\mathcal{L}_{CFM}(\theta) = \mathbb{E}_{t,q(x_1),p_t(x|x_1)}\|v_\theta(x, t) - u_t(x|x_1)\|^2 = \mathbb{E}_{t,q(x_1),p_t(x|x_1)}\|v_{\theta_{-b}}(x, t) + b - u_t(x|x_1)\|^2.$$

Note that the Jacobian matrix of $v_{\theta_{-b}}(x, t) + b$ reads as:

$$\mathbf{J}_v(\theta; x, t) := \left[\mathbf{J}_{v_{\theta_{-b}}}(\theta_{-b}; x, t), \mathbf{I}_{d_{out}}\right] \in \mathbb{R}^{d \times d_{out}}, \tag{D.1}$$

where $\mathbf{J}_{v_{\theta_{-b}}}(\theta_{-b}; x, t)$ is the Jacobian of the function $v_{\theta_{-b}}(x, t)$ w.r.t. $\theta_{-b}$. Since $d \geq d_{out}$, $\mathbf{J}_v(\theta; x, t)$ is full column-rank, i.e., $\mathbf{J}_v^\dagger(\theta; x, t)\mathbf{J}_v(\theta; x, t) = \mathbf{I}_{d_{out}}$.

Similar to the prove in (Lipman et al., 2022), to ensure existence of all integrals and to allow the changing of integration order (by Fubini's theorem) in the following, we assume that $q(x_1)$ and $p_t(x|x_1)$ are decreasing to zero at a sufficient speed as $\|x\| \to \infty$ and that $u_t, v_\theta, \nabla_\theta v_\theta$ are bounded. Then we have:

$$\nabla_\theta \mathcal{L}_{CFM}(\theta) = \nabla_\theta \mathbb{E}_{t,q(x_1),p_t(x|x_1)}\left\|v_{\theta_{-b}}(x, t) + b - u_t(x|x_1)\right\|^2$$

$$= \mathbb{E}_{t,q(x_1),p_t(x|x_1)}\left[\nabla_\theta\left\|v_{\theta_{-b}}(x, t) + b - u_t(x|x_1)\right\|^2\right]$$

$$= \mathbb{E}_{t,q(x_1),p_t(x|x_1)}\left[2\mathbf{J}_v(\theta; x, t)\left(v_{\theta_{-b}}(x, t) + b - u_t(x|x_1)\right)\right].$$

Therefore, we get

$$\mathcal{L}_{CFM}(\theta) = \mathbb{E}_{t,q(x_1),p_t(x|x_1)}\left\|v_{\theta_{-b}}(x, t) + b - u_t(x|x_1)\right\|^2$$

$$= \mathbb{E}_{t,q(x_1),p_t(x|x_1)}\left\|\mathbf{J}_v^\dagger(\theta; x, t)\mathbf{J}_v(\theta; x, t)\left(v_{\theta_{-b}}(x, t) + b - u_t(x|x_1)\right)\right\|^2$$

$$\leq \frac{1}{4}\mathbb{E}_{t,q(x_1),p_t(x|x_1)}\left\|2\mathbf{J}_v(\theta; x, t)\left(v_{\theta_{-b}}(x, t) + b - u_t(x|x_1)\right)\right\|^2$$

$$= \frac{1}{4}\left(\left\|\nabla_\theta\mathcal{L}_{CFM}(\theta)\right\|^2 + \text{Var}\left(\nabla_\theta\mathcal{L}_{CFM}(\theta; x_1, x, t)\right)\right),$$

which finish the proof. $\qquad\square$

**Lemma D.2** (Smoothness of $\mathcal{L}_{CFM}(\theta)$). Suppose Assumption 3.2 holds, then we have

$$\left\|\nabla_\theta^2\mathcal{L}_{CFM}(\theta)\right\| \leq L_\theta,$$

where $L_\theta < \infty$ is the constant that only depends on $\delta$, $d_{out}$, and $L$.

*Proof.* Given any $i \in [d_{out}]$, let $f_i(\theta, x, t) := (v_\theta(x, t) - u_t(x|x_1))_i$. We already have boundedness for the zeroth- and second-order, i.e.,

$$|f_i(\theta, x, t)| \leq \delta, \quad \left\|\nabla_\theta^2 f_i(\theta, x, t)\right\| \leq L,$$

then, by the Taylor's theorem with the Lagrange form of the remainder, we have

$$f_i(\theta_a, x, t) = f_i(\theta, x, t) + \nabla_\theta^\top f_i(\theta, x, t)(\theta_a - \theta) + \frac{1}{2}(\theta_a - \theta)^\top\nabla_\theta^2 f_i(\theta_\xi, x, t)(\theta_a - \theta), \quad \forall \theta_a, \theta \in \mathbb{R}^d,$$

where $\theta_\xi$ is some point between $\theta_a$ and $\theta$. Without loss of generality, we let $\|\theta_a - \theta\| = 1$, hence we can get

$$\left|\nabla_\theta^\top f_i(\theta, x, t)(\theta_a - \theta)\right| \leq |f_i(\theta_a, x, t) - f_i(\theta, x, t)| + \frac{1}{2}\left|(\theta_a - \theta)^\top\nabla_\theta^2 f_i(\theta_\xi, x, t)(\theta_a - \theta)\right| \leq 2\delta + \frac{L}{2}, \forall\theta_a, \theta.$$

Then we can get $\|\nabla_\theta f_i(\theta, x, t)\| \leq 2\delta + \frac{L}{2}$. Then we have

$$\|\mathbf{J}_v(\theta; x, t)\| = \left\|\begin{bmatrix}\nabla_\theta f_1(\theta, x, t) \\ \vdots \\ \nabla_\theta f_{d_{out}}(\theta, x, t)\end{bmatrix}\right\| \leq 2\delta + \frac{L}{2}.$$

Now, we are ready to prove the smoothness of $\mathcal{L}_{CFM}(\theta)$. The Hessian of the $\mathcal{L}_{CFM}(\theta)$ is

$$\nabla_\theta^2\mathcal{L}_{CFM}(\theta) = \nabla_\theta^2\mathbb{E}_{t,q(x_1),p_t(x|x_1)}\left\|v_{\theta_{-b}}(x, t) + b - u_t(x|x_1)\right\|^2$$

$$= \mathbb{E}_{t,q(x_1),p_t(x|x_1)}\left[\nabla_\theta^2\left\|v_{\theta_{-b}}(x, t) + b - u_t(x|x_1)\right\|^2\right]$$

$$= 2\mathbb{E}_{t,q(x_1),p_t(x|x_1)}\left[\mathbf{J}_v\mathbf{J}_v^\top(\theta; x, t) + \sum_{i=1}^{d_{out}}(v_\theta(x, t) - u_t(x|x_1))_i\nabla^2(v_\theta(x, t))_i\right].$$

Note that, we already have

$$\left\|(v_\theta(x,t) - u_t(x|x_1))_i \nabla^2(v_\theta(x,t))_i\right\| \le \delta L.$$

Then, by Jensen inequality, we can conclude that

$$\left\|\nabla_\theta^2 \mathcal{L}_{CFM}(\theta)\right\| = 2\left\|\mathbb{E}_{t,q(x_1),p_t(x|x_1)}\left[\mathbf{J}_v\mathbf{J}_v^\top(\theta;x,t) + \sum_{i=1}^{d_{out}}(v_\theta(x,t) - u_t(x|x_1))_i \nabla^2(v_\theta(x,t))_i\right]\right\|$$

$$\le 2\mathbb{E}_{t,q(x_1),p_t(x|x_1)}\left\|\mathbf{J}_v\mathbf{J}_v^\top(\theta;x,t) + \sum_{i=1}^{d_{out}}(v_\theta(x,t) - u_t(x|x_1))_i \nabla^2(v_\theta(x,t))_i\right\|$$

$$\le 2\mathbb{E}\left\|\mathbf{J}_v\mathbf{J}_v^\top(\theta;x,t)\right\| + 2\mathbb{E}\left\|\sum_{i=1}^{d_{out}}(v_\theta(x,t) - u_t(x|x_1))_i \nabla^2(v_\theta(x,t))_i\right\|$$

$$\le 2\left(2\delta + \frac{L}{2}\right)^2 + 2d_{out}\delta L := L_\theta.$$

$\square$

*Proof of Theorem. 3.3.* Due the the $L_\theta$-smoothness from Lemma D.2 and together with the update rule of SGD, we have

$$\mathcal{L}_{CFM}(\theta^{k+1}) \le \mathcal{L}_{CFM}(\theta^k) + \left\langle\nabla_\theta\mathcal{L}_{CFM}(\theta^k), \theta^{k+1} - \theta^k\right\rangle + \frac{L_\theta}{2}\left\|\theta^{k+1} - \theta^k\right\|^2$$

$$= \mathcal{L}_{CFM}(\theta^k) - \eta\left\langle\nabla_\theta\mathcal{L}_{CFM}(\theta^k), \nabla_\theta\mathcal{L}_{CFM}(\theta^k;x_1,x,t)\right\rangle + \frac{L_\theta\eta^2}{2}\left\|\nabla_\theta\mathcal{L}_{CFM}(\theta^k;x_1,x,t)\right\|^2.$$

By taking expectation conditioned on $\theta^k$, and denote $V := \mathrm{Var}\left(\nabla_\theta\mathcal{L}_{CFM}(\theta;x_1,x,t)\right)$, we can get

$$\mathbb{E}\left[\mathcal{L}_{CFM}(\theta^{k+1})\right] \le \mathcal{L}_{CFM}(\theta^k) - \eta\left\|\nabla_\theta\mathcal{L}_{CFM}(\theta^k)\right\|^2 + \frac{L_\theta\eta^2}{2}\mathbb{E}\left\|\nabla_\theta\mathcal{L}_{CFM}(\theta^k;x_1,x,t)\right\|^2$$

$$\le \mathcal{L}_{CFM}(\theta^k) - \eta\left\|\nabla_\theta\mathcal{L}_{CFM}(\theta^k)\right\|^2 + \frac{L_\theta\eta^2}{2}\left\|\nabla_\theta\mathcal{L}_{CFM}(\theta)\right\|^2 + \frac{L_\theta\eta^2 V}{2}$$

$$\overset{(a)}{\le} \mathcal{L}_{CFM}(\theta^k) - \frac{\eta}{2}\left\|\nabla_\theta\mathcal{L}_{CFM}(\theta^k)\right\|^2 + \frac{L_\theta\eta^2 V}{2}$$

$$\overset{(b)}{\le} \mathcal{L}_{CFM}(\theta^k) + \frac{\eta}{2}\left(V - 4\mathcal{L}_{CFM}(\theta^k)\right) + \frac{L_\theta\eta^2 V}{2}$$

$$\le (1 - 2\eta)\mathcal{L}_{CFM}(\theta^k) + \eta V$$

$$\le (1 - 2\eta)^k\mathcal{L}_{CFM}(\theta^0) + \frac{V}{2}$$

$$\overset{(c)}{\le} (1 - 2\eta)^k\mathcal{L}_{CFM}(\theta^0) + 2d\mathbb{E}_{t,p_t(x)}[\|\nabla_\theta v_\theta(x,t)\|^2]\int h_t^2\left(1 - \mathcal{S}\left(\frac{a_t}{m_t}\right)\right)\mathrm{d}t$$

$$\le (1 - 2\eta)^k\mathcal{L}_{CFM}(\theta^0) + 2d\left(2\delta + \frac{L}{2}\right)^2\int h_t^2\left(1 - \mathcal{S}\left(\frac{a_t}{m_t}\right)\right)\mathrm{d}t,$$

where (a) is due to $\eta \le \frac{1}{L_\theta}$, (b) comes from Lemma D.1, and (c) comes from Lemma 3.1. $\square$

Theorem 3.3 shows that during training, the loss decreases rapidly until the model converges to a local neighborhood around the global minimizer. Within this neighborhood, the training loss is proportional to both the peak variance scale of the sampled regions under the sampling policy and the dimensionality of the data (or, the latent space dimensionality).

As a result, for the same reduction in peak variance scale achieved by VR-Sampling, the impact on training loss becomes more significant with higher-dimensional latents. This could explain why the improvements are more noticeable in higher-resolution images or latents in our experimental results.

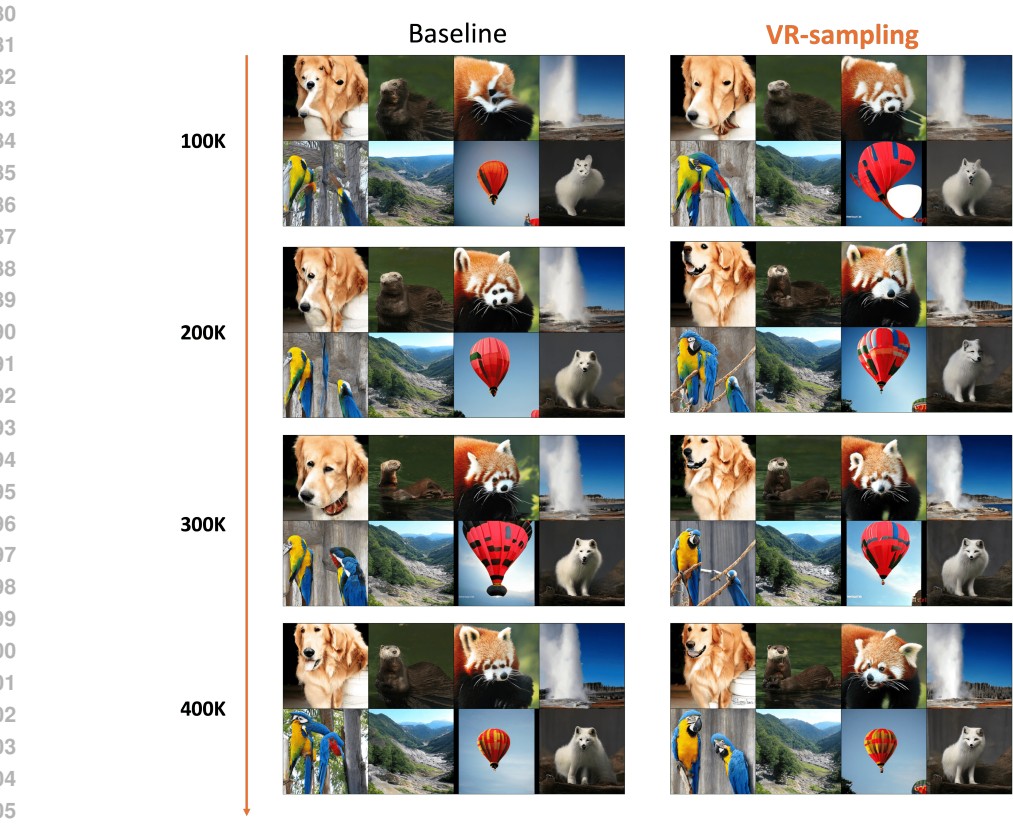

Figure 13: Linear noise scheduler in ImageNet $512 \times 512$ (zoom in to observe details).

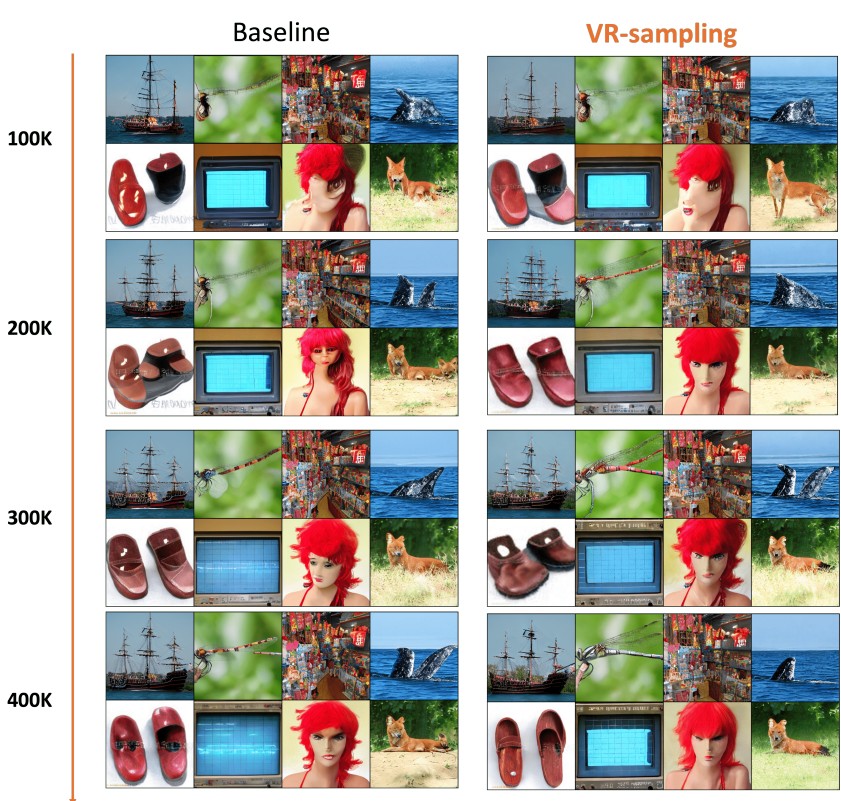

Figure 14: Cosine noise scheduler in ImageNet $256 \times 256$ (zoom in to observe details).

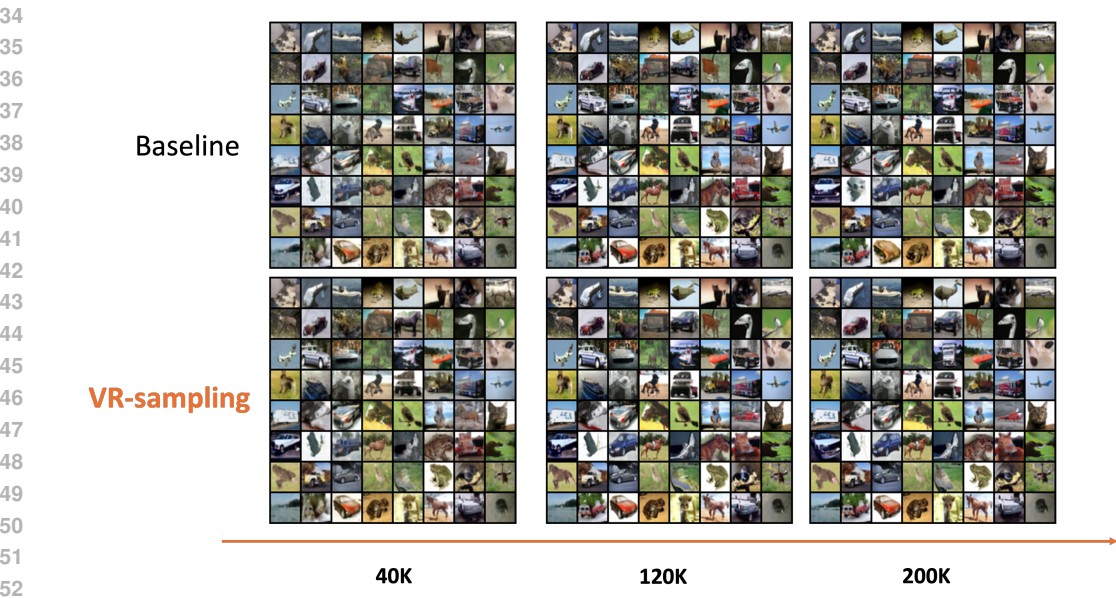

Figure 15: Diffusion noise scheduler for CIFAR10

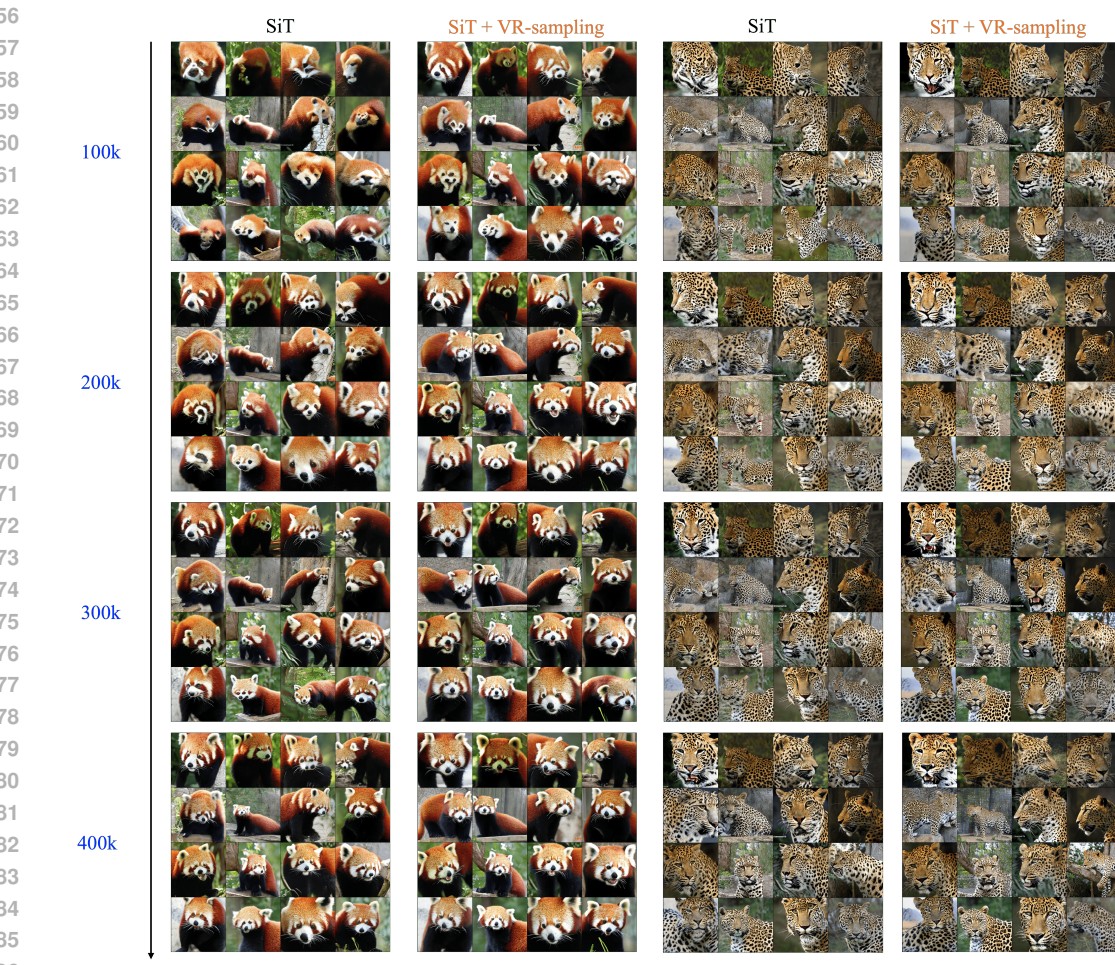

Figure 16: More qualitative results on linear noise scheduler in ImageNet $512 \times 512$

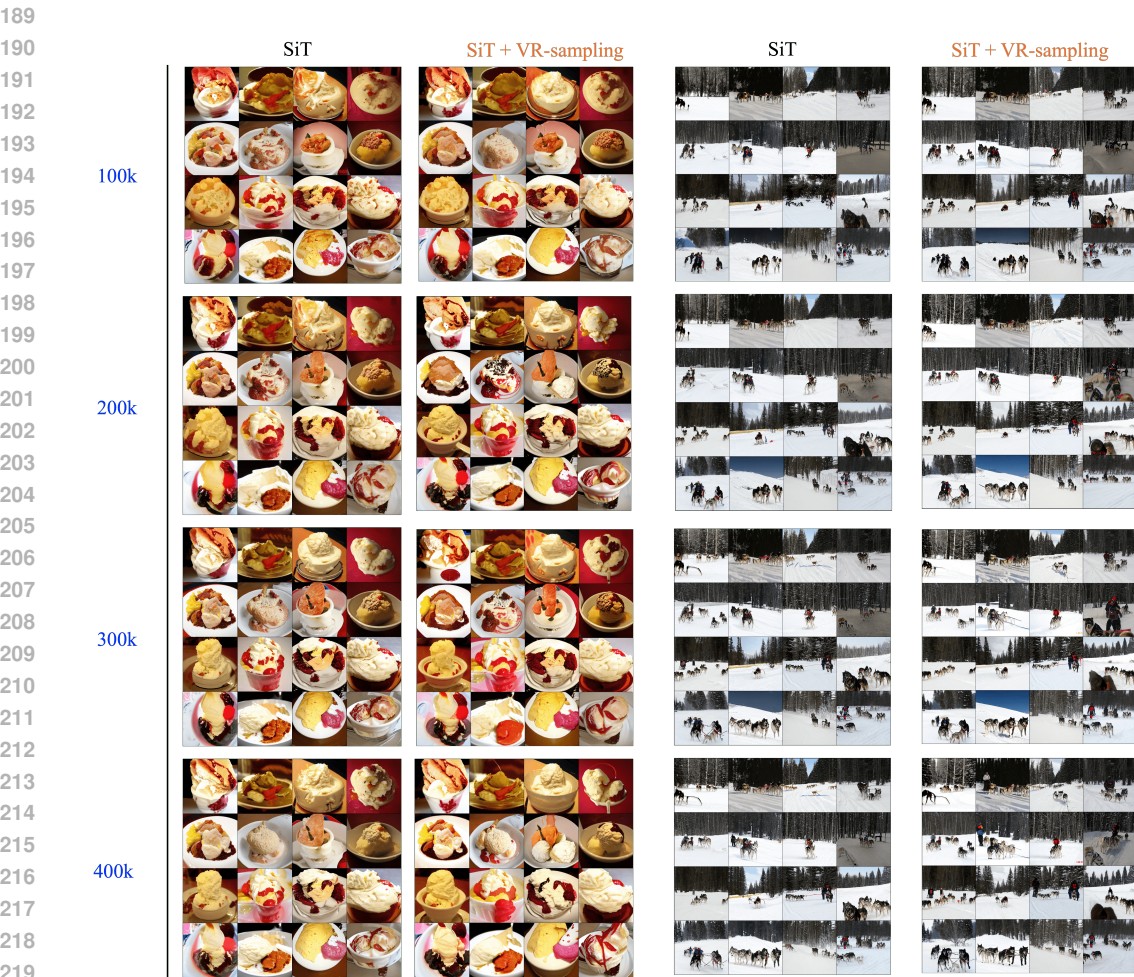

Figure 17: More qualitative results on linear noise scheduler in ImageNet $512 \times 512$

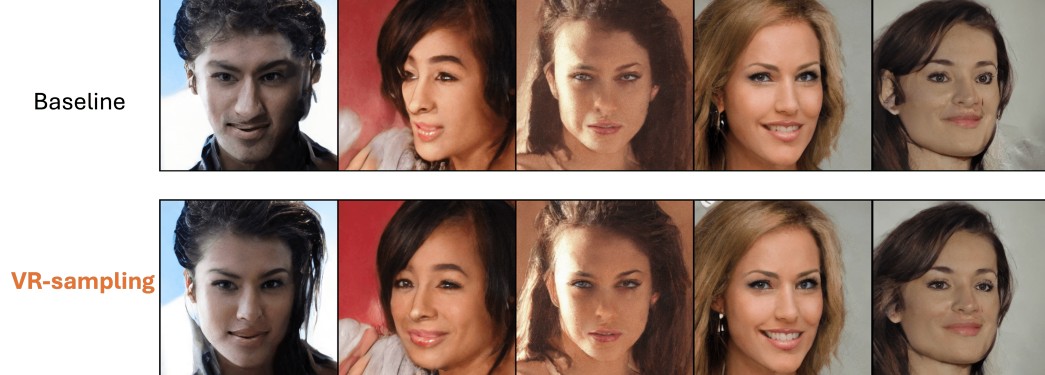

Figure 18: Qualitative results on CelebA datasets under linear noise scheduler at 40k iterations with baseline FID 23.5 and VR-sampling FID as 22.8.

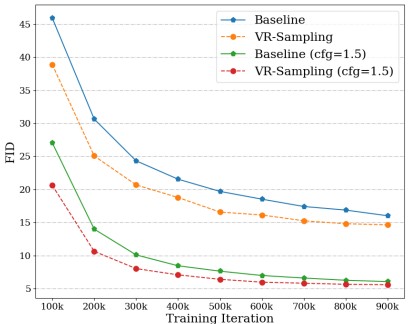

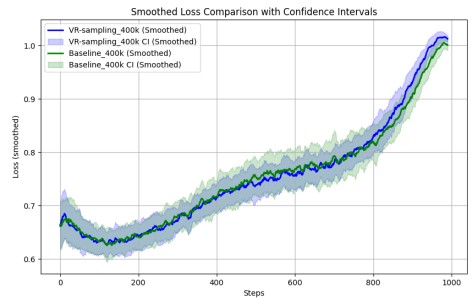

Figure 19: FID values with more training iterations.

Figure 20: Loss curves of linear noise scheduler under ImageNet-256 at 400K iterations.

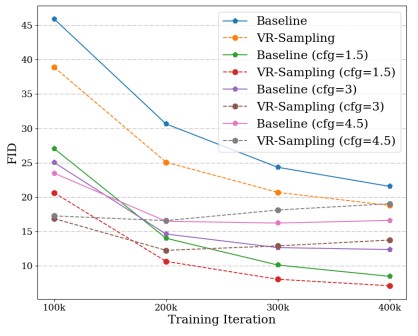

(a) FID under cfg=3.0 and cfg=4.5.

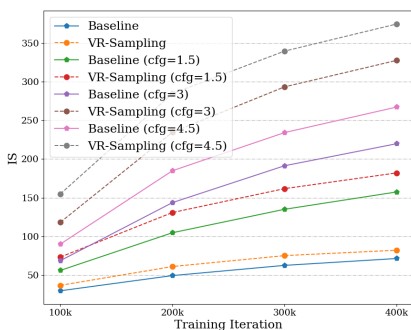

(b) IS under cfg=3.0 and cfg=4.5.

Figure 21: FID and IS curves for higher cfg values under linear noise schedulers for ImageNet256 datasets.

