# OpenReview forum: "VR-Sampling: Accelerating Flow Generative Model Training with Variance Reduction Sampling"
_ICLR.cc/2025/Conference — ICLR 2025 Conference Withdrawn Submission_

### Official Review · Reviewer_HTm4 · 2024-10-27

**Soundness:** 3
**Presentation:** 3
**Contribution:** 3
**Rating:** 8
**Confidence:** 3

**Summary:**

This paper proposes a Variance-Reduction Sampling (VR-sampling) strategy that samples the time steps in high-variance region more frequently to enhance training efficiency in diffusion models. The method first identifies the root cause of the high variance and proposes an effective solution for importance sampling. The proposed strategy is also supported with proof of bounds and convergence analysis. The results show that the proposed method outperforms various baselines, experimented on ImageNet and CIFAR datasets and different model architectures including U-Net, DiT and SiT.

**Strengths:**

- How to sample time steps is indeed an important problem in traininig diffusion models, the method is with a clear motivation.
- The paper is well-written and easy to follow.
- The results including FID curves and tables clearly and consistently show the effectiveness of the proposed method on different variants of flow models and model architectures, which are widely used in generative modeling recently.
- The proposed method is also supported with necessary proof of bounds and convergence analysis.
- It is appreciated that the paper includes an anonymous code link for reproducibility.

**Weaknesses:**

- The paper might lack some dicussions to related method in diffusion models. For example, in line 196, Iterative α-(de)Blending [1] also shares the same formulation as rectified flow. In Monte Carlo sampling, variance reduction can be achieved by using not just importance sampling, but also control variates and correlated sampling, which are initially explored [2,3] for diffusion models. Also, there exists similar line of work considering variance reduction in deep learning such as Deep Learning with Importance Sampling [4] (and there are more), which might also be worth mentioning.
- The visual improvements of VR-sampling shown in Figure 14, 15 seem not significant to me. It will be great to have results on one of the Animal/Human faces (e.g., AFHQ, CelebA, FFHQ), or one of the LSUN datasets. As observed in Figure 13 and 14, the method seems to be working better on face images. Also, results on LSUN bedroom or church might be interesting as the images are with specific textures and patterns.


[1] Iterative 𝛼-(de)Blending: a Minimalist Deterministic Diffusion Model, Heitz et al. 2023

[2] Variance reduction of diffusion model's gradients with Taylor approximation-based control variate, Jeha et al. 2024

[3] Blue noise for diffusion models, Huang et al. 2024

[4] Not All Samples Are Created Equal: Deep Learning with Importance Sampling, Katharopoulos and Fleuret 2018

**Questions:**

- Is it right that the proposed VR-sampling is only performed in the training phase, not in the inference phase?
- VR-sampling can support different choice of Noise Scheduler (Diffusion, Linear, Cosine), but it seems currently only tested on deterministic ODE formulation. Is it possible to extend it to SDE formulation (as a future work)?
- Does Figure 2 show that S is highly correlated with the resolution, as both CIFAR and ImageNet 256 are using 64x64 resolution?
- Could the author provide some insights how the cfg value can affect the results? I can see with higher cfg in Figure 3, 4, 5, the FID curve for VR-sampling can converge more similar to the baseline (Fig. 3). That said, what would you expect to happen if cfg is higher like 4.0, would the method still clearly outperform the baseline?
- In line 324-331, the author mentioned VR-sampling takes extra time to find the probability density function (PDF) and construct $x_t$. To my understanding, this process requires evaluating the equation (missing equation number?) in line 290 and the term $p_t(x|x_1,i)$ using the neural network prediction. But I am not clear which state/checkpoint of the neural network is used to compute this term?
- It is briefly mentioned "Perception prioritized training of diffusion models" use heuristics to sample time steps based on signal-to-noise ratio (SNR). But I am wondering how it compares to the proposed VR-sampling method?
- The improvement seems more visible in Figure 13, than the improvement shown in Figure 14, 15. Does it give a hint that VR-sampling might work better with higher reoslution images or latents?

Some minor comments:
- I would suggest Figure 13-15 can be larger to better see more details and utilize the space.
- In table 1 "Ho et al., 2020b" is the same as "Ho et al., 2020a". It is cited twice.
- In table 6, how CFG scale is used in training? According to "Classifier-free diffusion guidance" [Ho and Salimans 2022], the training does not use the value directly, but with a random discard of the conditioning to train unconditionally.

---

> ### Author Response · Authors · 2024-11-23
> **Reply to Reviewer HTm4**
>
> >W1. The paper might lack some dicussions to related method in diffusion models.
>
> **Reply:** Thanks for your advice. We have included the following discussion in the revised version of the paper in related works:
>
> Several works are related to deterministic generative models. For instance, [3] proposes incorporating correlated noise instead of pure Gaussian noise into deterministic diffusion models, which improves image quality. Similarly, [1] introduces iterative $\alpha$-blending, which blends and deblends samples between two densities, a concept that shares similarities with SiT [5], which connects two distributions through an interpolant framework.
>
> Our work builds on the idea of importance sampling to reduce gradient variance. For example, [4] demonstrates that importance sampling can focus computation on informative examples, reduce gradient variance during training, and accelerate convergence. Furthermore, [2] explores variance reduction using a $k$-th order Taylor expansion on the training objective and its gradient. In contrast, our approach directly targets high-variance regions during training, sampling them more frequently to reduce variance and improve efficiency, without requiring approximations like those in [2].
>
>
>
> >W2. The visual improvements of VR-sampling shown in Figure 14, 15 seem not significant to me.
>
> **Reply:** Thank you for your valuable suggestion. In the revised paper, we have included additional qualitative results on the ImageNet 512×512 dataset, which are presented in Figures 16–17 in the Appendix. Additionally, we conducted experiments on the CelebA dataset, and the corresponding qualitative results can be found in Figure 18 in the Appendix. These results demonstrate that, within the same number of training iterations, our method produces face with finer and better details. Due to the limited rebuttal period, we were unable to train on LSUN datasets. We will try to include more results in the final version.
>
> [1] Iterative 𝛼-(de)Blending: a Minimalist Deterministic Diffusion Model, Heitz et al. 2023
>
> [2] Variance reduction of diffusion model's gradients with Taylor approximation-based control variate, Jeha et al. 2024
>
> [3] Blue noise for diffusion models, Huang et al. 2024
>
> [4] Not All Samples Are Created Equal: Deep Learning with Importance Sampling, Katharopoulos and Fleuret 2018
>
> [5] Sit: Exploring flow and diffusion-based generative models with scalable interpolant transformers. 2024
>
> >Q1. Is it right that the proposed VR-sampling is only performed in the training phase, not in the inference phase?
>
> **Reply:** Our VR-sampling strategy is specifically designed based on an analysis of gradient variance during training optimization. Its primary goal is to balance gradient variance and accelerate the training process, where it demonstrates significant improvements.
>
> For inference, we conducted a simple evaluation, but the performance gains were minimal. For example, in the SiT XL/2 network with a linear noise scheduler, the baseline achieves an FID of 15.92, while applying the VR-sampling strategy during inference results in an FID of 15.89. We hypothesize that in the inference phase, the intermediate steps also play a critical role, but the sampling strategy requires a different approach to be effective. This is an interesting direction for further research, and we plan to explore it in future work.
>
> >Q2. VR-sampling can support different choice of Noise Scheduler (Diffusion, Linear, Cosine), but it seems currently only tested on deterministic ODE formulation. Is it possible to extend it to SDE formulation (as a future work)?
>
> **Reply:** We chose SiT [5] as our baseline because its interpolant formulation is widely used in large text-to-image (T2I) and text-to-video (T2V) models, such as Stable Diffusion 3 and OpenSora. SiT is specifically built on the ODE formulation of the generative process. Extending our method to the SDE formulation is a promising direction for future work. In the SDE setting, it would require evaluating the gradient variance of the score matching loss, which presents additional challenges and opportunities for further exploration.
>
> >Q3. Does Figure 2 show that S is highly correlated with the resolution, as both CIFAR and ImageNet 256 are using 64x64 resolution?
>
> **Reply:** In Figure 2, CIFAR is trained in pixel space with dimensions of **3×32×32**, while ImageNet 256 is trained in the latent space with dimensions of **4×32×32**. Due to these differences in dimensionality, their respective $\mathcal{S}$ curves are distinct.

---

> > ### Author Response · Authors · 2024-11-23
> > **Reply to Reviewer HTm4**
> >
> > >Q4. Could the author provide some insights how the cfg value can affect the results?
> >
> > **Reply:** We add the FID curves under cfg=3.0 and cfg=4.5 under linear noise schedulers for ImageNet256 datasets. See it in the newly updated Fig. 21 in Appendix. A high classifier-free guidance (cfg) scale introduces a trade-off between fidelity (alignment to conditional input) and diversity/realism. Specifically:
> > - Inception Score (IS) improves with higher cfg values as the generated images become more distinct and coherent with their class features, resulting in higher confidence predictions.
> > - FID, however, often suffers at high cfg values because the generated images deviate from the natural statistics of the dataset due to overfitting to the class guidance.
> >
> > In Fig. 21 (a), for higher cfg values (e.g., cfg=3.0 and cfg=4.5), the trends of the FID curves differ from those at lower cfg values (e.g., cfg=1.0 and cfg=1.5). Initially, our method outperforms the baseline in terms of FID, but with longer training, the FID eventually becomes worse than the baseline. This occurs because the model fits the data better but becomes overly aligned to the class conditions at higher cfg values, reducing diversity and realism.
> >
> > However, as shown in Fig. 21 (b), the Inception Scores (IS) consistently outperform the baseline methods across all cfg values, indicating better semantic alignment and class fidelity. Additionally, at higher cfg scales, the differences between our method and the baseline are more pronounced, highlighting the advantages of VR-sampling in these settings.
> >
> > >Q5. In line 324-331, the author mentioned VR-sampling takes extra time to find the probability density function (PDF) and construct. But I am not clear which state/checkpoint of the neural network is used to compute this term?
> >
> > **Reply:** Thanks. We have labeled the equation to Eq. (3.3) in 290. Actually, the simulation of equation on line 290 is not related with neural networks. The calculation of Eq. (3.3) is shown as follows:
> >
> > We sample $M$ data points $(x_{1,i})^M_{i=1}$ and sample $K$ pure Gaussian points $(\epsilon_l)_{l=1}^K$. Then
> >
> > \begin{align}
> > \mathcal{S}\left(\frac{a_t}{m_t}\right) &= \frac{1}{Md}\sum_{i=1}^M E_{p_t(x|x_{1,i})}  || \sum_{j=1}^M x_{1,j} p_t(x_{1,i}|x)||^2
> > = \frac{1}{Md} \sum_{l=1}^K \sum_{i=1}^M || \sum_{j=1}^M x_{1,j} p_t(x_{1, j}|x_{1,i}+m_t/a_t \epsilon_l)||^2,
> > \end{align}
> > where according to Bayes' rule, $$p_t(x_{1,i}|x)=\frac{p_t(x|x_{1,i})q(x_{1,i})}{\sum_{j=1}^M q(x_{1,j})p_t(x|x_{1,j})} = \frac{e^{-\frac{a_t^2}{2m_t^2}\|x_i - x\|^2}}{\sum_{j=1}^M e^{-\frac{a_t^2}{2m_t^2}\|x_j - x\|^2}}.$$
> >
> >
> > >Q6.  But I am wondering how it compares to the proposed VR-sampling method?
> >
> > **Reply:** In our paper, we compared our VR-sampling method with the SoTA method--SpeeD [6], in the diffusion noise scheduler setting, as shown in Fig. 7. Our method demonstrates faster convergence than SpeeD. According to [6], their approach outperforms “Perception Prioritized Training of Diffusion Models,” particularly during the early training stages. Therefore, we focused our comparison on the SoTA method and did not repeat experiments with the perception-prioritized training approach.
> >
> > [6] A closer look at time steps is worthy of triple speed-up for diffusion model training. 2024.
> >
> > >Q7. Does it give a hint that VR-sampling might work better with higher reoslution images or latents?
> >
> > **Reply:** To some extent, this observation is accurate. According to our theoretical analysis (Theorem 3.3), during training, the loss decreases rapidly until the model converges to a local neighborhood around the global minimizer. Within this neighborhood, the training loss is proportional to both the peak variance scale of the sampled regions under the sampling policy and the dimensionality of the data (or, the latent space dimensionality).
> >
> > As a result, for the same reduction in peak variance scale achieved by VR-Sampling, the impact on training loss becomes more significant with higher-dimensional latents. This could explain why the improvements are more noticeable in higher-resolution images or latents.
> >
> > Minor:
> > >1. I would suggest Figure 13-15 can be larger to better see more details and utilize the space.
> >
> > **Reply:** Thanks. We make it larger and add more qualitative clearer and larger results in Fig.16-17 in Appendix.
> >
> > >2. In table 1 "Ho et al., 2020b" is the same as "Ho et al., 2020a". It is cited twice.
> >
> > **Reply:** Thanks. We have corrected it.
> >
> > >3. In table 6, how CFG scale is used in training?
> >
> > **Reply:** Here we follow the setting of SiT [5] and during training, cfg=4.0 is just a value greater than 1.0 to indicate training in classifier-free guidance (see the line 222 in train.py in the officially released code of SiT [7]).
> >
> > [7] https://github.com/willisma/SiT/blob/main/train.py

---

> > > ### Comment · Reviewer_HTm4 · 2024-11-23
> > >
> > > I would like to thank the authors for adding new discussions, experiments, as well as updating the paper.
> > >
> > > - Thanks for the clarifications on Q3, Q4, Q6, Q7, Minor. For Q4, it is interesting to see the Inception Scores (IS) is consistently better with VR-sampling. Q7 seems an interesting point to mention in the paper.
> > >
> > > - For W2, the CelebA experiment in Figure 18 shows similar quality for both methods. The improvements on Figure 16 look more obvious to me. Given the time consumption of VR-Sampling is only slightly higher, the proposed method can be valuable for high-resolution images.
> > >
> > > - For Q5, does it mean Eq. 3.3 relies on the magnitude of noise and SNR ($\alpha_t^2$ / $m_t^2$)?

---

> > > > ### Author Response · Authors · 2024-11-24
> > > > **Reply to Reviewer HTm4**
> > > >
> > > > Thanks for your feedback.
> > > >
> > > > - We will add the analysis of Q7 in the paper.
> > > >
> > > > - For Q5, yes.  This is the main point of our paper. SiT unifies the rectified flow loss and the previous diffusion loss within the interpolant framework. The primary difference between these losses lies in the noise schedulers, specifically the variation of the signal-to-noise ratio w.r.t  $t $. Importantly, the design of VR-sampling is adaptable to different noise schedulers, further demonstrating its general applicability.

---

> > > > > ### Comment · Reviewer_HTm4 · 2024-11-24
> > > > >
> > > > > Thank you for addressing most of my concerns. I will increase my score. One more comment, if you are going to include CelebA and more results like LSUN, please also show the FID or related scores.

---

> > > > > > ### Author Response · Authors · 2024-11-25
> > > > > > **Reply to Reviewer HTm4**
> > > > > >
> > > > > > Thanks very much for your helpful advice.
> > > > > >
> > > > > > I will add the related results of LSUN bedroom before the rebuttal deadline.

---

### Official Review · Reviewer_Ud3A · 2024-10-31

**Soundness:** 3
**Presentation:** 3
**Contribution:** 2
**Rating:** 5
**Confidence:** 3

**Summary:**

The paper introduces a Variance Reduction-based diffusion timesteps sampling strategy to accelerate the training of DMs. The authors identify that the variance of gradient estimates in the training process is higher at intermediate timesteps, which affects training stability. They propose VR-sampling to prioritize sampling from high-variance regions, thereby accelerating training. The method is shown to significantly speed up training across various noise schedulers and data dimensions.

**Strengths:**

* This papers provides theoretical analysis on gradient variance for conditional flow matching loss and proves the convergence rate during training.
* The experimental results are clear and well verified the effectiveness of the VR-reduction sampling strategy.

**Weaknesses:**

1. The paper provides a theoretical analysis of the variance of the DSM loss and the convergence speed of DMs training in SGD. Based on this analysis, the paper proposes a method to accelerate training of DMs. While the experimental results appear promising, there is a lack of analysis explaining why the VR-Sampling strategy, specifically "sampling timesteps in high-variance regions more frequently," leads to improved convergence speed.

2. Building on point 1, it seems that the proposed method is anagolous to some empirical findings, such as the Log-normal and logit-normal techniques mentioned in [1] and [2]. The relationship between the theoretical insights and the proposed algorithm could be made more explicit.

3. Section 4.2 is not entirely convincing. The results on the baseline models are not as strong as those reported in the original paper. For instance, in [3], the FID score (cfg=1.5) is 2.06 and 2.62 respectively on ImageNet 256 and ImageNet 512. In contrast, the best reported FID score for the baseline in this paper is 8.34, as shown in Table 2. While I understand that the experiments have controlled the number of training iterations, I believe a comparison of the convergence points is also necessary to provide a more comprehensive evaluation.

4. On line 290, the Monte Carlo approximation is somewhat confusing. Are you using samples $x_1\sim q(x_1)$ to compute both the outer expectation and inner expectation? Could the authors please provide a more detailed explanation to clarify this concept?

**minor points**
* Equation at line 290 is not labeled.

[1] Tero Karras, et al. Elucidating the design space of diffusion- based generative models.

[2] Patrick Esser, et al. Scaling rectified flow transformers for high-resolution image synthesis. In ICML 2024.

[3] Nanye Ma, et al. Sit: Exploring flow and diffusion-based generative models with scalable interpolant transformers.

**Questions:**

* Could the authors explain more detailedly about the VR-sampling strategy (line 315~323)?
* Could the authors offer a figure like FIgure.5(a) in [1] to show the results on how the VR-sampling strategy improve the gradient variance?

[1] Tero Karras, et al. Elucidating the design space of diffusion- based generative models.

---

> ### Author Response · Authors · 2024-11-23
> **Reply to Reviewer Ud3A**
>
> >W1. The paper provides a theoretical analysis of the variance of the DSM loss and the convergence speed of DMs training in SGD. Based on this analysis, the paper proposes a method to accelerate training of DMs. While the experimental results appear promising, there is a lack of analysis explaining why the VR-Sampling strategy, specifically "sampling timesteps in high-variance regions more frequently," leads to improved convergence speed.
>
> **Reply:** The theoretical upper bound from Theorem 3.3 shows that training consists of two phases. In the first phase, the training loss decreases rapidly, converging to a local neighborhood around the global minimizer. In the second phase, the model weights fluctuate within this neighborhood. However, convergence slows significantly in the second phase.
>
> The size of this local neighborhood is proportional to the  variance scale under the sampling policy. By reducing this variance scale, our VR-Sampling strategy allows the model to spend more time in the first phase with the rapid convergence. This enables faster convergence and leads to higher-quality solutions compared to conventional sampling strategies.
>
> >W2. Building on point 1, it seems that the proposed method is anagolous to some empirical findings, such as the Log-normal and logit-normal techniques mentioned in [1] and [2]. The relationship between the theoretical insights and the proposed algorithm could be made more explicit.
>
> **Reply:**   In [1], the authors observe that different levels of loss occur at various noise levels, with significant reductions being possible primarily at intermediate noise levels. This observation motivates their use of log-normal sampling to emphasize training efforts on these intermediate noise levels. Similarly, [2] employs logit-normal sampling to focus training efforts on intermediate timesteps. While both works intuitively highlight the importance of intermediate timesteps, our work delves deeper into the training optimization process and reveals that high-variance regions during training influence convergence.
>
> Building on our theoretical analysis, we propose a sampling strategy aimed at reducing gradient variance. Compared to the approaches in [1] and [2], our VR-sampling more precisely identifies high-variance regions and is generalizable across different noise schedulers and data dimensions, resulting in better acceleration performance. This is also demonstrated in Section 4.3, where aligning the density of logit-normal sampling with our simulated density improves training speed by 38% compared to the commonly used default setting (m=0).
>
> >W3. Section 4.2 is not entirely convincing....While I understand that the experiments have controlled the number of training iterations, I believe a comparison of the convergence points is also necessary to provide a more comprehensive evaluation.
>
> **Reply:**  We acknowledge the importance of evaluating results at convergence points. SiT achieves its state-of-the-art FID at approximately **7000K** iterations. However, due to the limited rebuttal period, we were unable to reach this iteration count, as it requires nearly 42 days of training on 8×A100 GPUs. We have included additional experimental results showing FID values at approximately **950K** iterations (which required 6.5 days of training) for ImageNet-256 under the linear noise scheduler in SiT XL/2. Please refer to the newly added FID curves in Fig. 19 in the Appendix for more details. These results indicate that the SiT baseline achieves an FID of 16.0 (cfg=1) and 6.0 **(cfg=1.5)** at 950K iterations, whereas our method reaches a comparable FID of 16.1 (cfg=1) and 5.99 **(cfg=1.5)** at just 600K iterations, highlighting the efficiency of our approach. We will try to include the results around convergence points in the final version of the paper.
>
> The primary reason we report results at 400K iterations is that our method is **generally applicable to any noise scheduler (including both diffusion and flow model settings) and data dimension**. To validate its effectiveness experimentally, we conducted at least 9 experiments across various settings. Running all these experiments to full convergence (around **7000K** iterations in ImageNet-256) would be highly resource-intensive. For instance, a single experiment on ImageNet256 at convergence would take over one month using 8×A100 GPUs, which is infeasible given our limited computational resources. Therefore, we report FID values at 400K iterations for ImageNet-256 and 512, a practical and consistent choice that aligns with the settings used in comparable works, such as [5,6]. This approach ensures a fair and resource-efficient evaluation of our method.

---

> ### Author Response · Authors · 2024-11-23
> **Reply to Reviewer Ud3A**
>
> >W4. On line 290, the Monte Carlo approximation is somewhat confusing. Are you using samples to compute both the outer expectation and inner expectation? Could the authors please provide a more detailed explanation to clarify this concept?
>
> **Reply:**   Yes. For the outer and inner expectation, we both use the samples. We sample $M$ data points $(x_{1,i})^M_{i=1}$ and sample $K$ pure Gaussian points $(\epsilon_l)_{l=1}^K$. Then
>
> \begin{align}
> \mathcal{S}\left(\frac{a_t}{m_t}\right) &= \frac{1}{Md}\sum_{i=1}^M E_{p_t(x|x_{1,i})} || \sum_{j=1}^M x_{1,j} p_t(x_{1,i}|x)||^2
> = \frac{1}{Md} \sum_{l=1}^K \sum_{i=1}^M || \sum_{j=1}^M x_{1,j} p_t(x_{1, j}|x_{1,i}+m_t/a_t \epsilon_l)||^2,
> \end{align}
> where according to Bayes' rule, $$p_t(x_{1,i}|x)=\frac{p_t(x|x_{1,i})q(x_{1,i})}{\sum_{j=1}^M q(x_{1,j})p_t(x|x_{1,j})} = \frac{e^{-\frac{a_t^2}{2m_t^2}|x_i - x|^2}}{\sum_{j=1}^M e^{-\frac{a_t^2}{2m_t^2}|x_j - x|^2}}.$$
>
> This calculation process is also shown in [4].
>
> >W5. Equation at line 290 is not labeled.
>
> **Reply:** Thanks. We have added the label.
>
> [1] Tero Karras, et al. Elucidating the design space of diffusion- based generative models.
>
> [2] Patrick Esser, et al. Scaling rectified flow transformers for high-resolution image synthesis. In ICML 2024.
>
> [3] Nanye Ma, et al. Sit: Exploring flow and diffusion-based generative models with scalable interpolant transformers.
>
> [4] Neta Shaul, et al. On kinetic optimal probability paths for generative models.
>
> [5] A closer look at time steps is worthy of triple speed-up for diffusion model training. 2024.
>
> [6] Towards faster training of diffusion models: An inspiration of a consistency phenomenon. 2024.
>
> Questions:
> >Q1. Could the authors explain more detailedly about the VR-sampling strategy (line 315~323)?
>
> **Reply:**  Thank you for pointing this out. We have clarified the sampling process in the revised paper as follows:
>
> After using Monte Carlo simulations to obtain variance curves ranging from 0 to 1, these curves are typically non-normalized and cannot directly serve as valid probability density functions (PDFs). To address this,
>
> (1) We first normalize the simulated curve by calculating the total area under the curve (via integration) and dividing the curve by the total area.
>
> (2) We compute the cumulative sum of the normalized curve to derive the cumulative distribution function (CDF) $F(t)$.
>
> (3) Using the CDF, we construct an interpolation function to approximate its inverse $F^{-1}(u), u\sim Uniform(0,1)$.
>
> (4) The inverse CDF maps a uniformly distributed random variable to the domain of $t$, ensuring that the resulting samples follow the distribution defined by the probability density function (PDF) $\pi(t)$, where $\pi(t) = F'(t)$.
>
> Here the inverse CDF serves as a crucial tool for sampling from the desired density. Our final simulated sampling density is $\pi(t)$, and we provide visualizations of its simulated distribution in Fig. 1 (d)-(f) for better understanding.
>
>
> >Q2. Could the authors offer a figure like FIgure.5(a) in [1] to show the results on how the VR-sampling strategy improve the gradient variance?
>
> **Reply:** We show the loss curves of baseline method and our VR-sampling strategy under linear noise scheduler Fig.20 in Appendix. We have some interesting observations: 1. The shapes of the loss curves under the linear noise scheduler (i.e., rectified flow form) differ significantly from the diffusion loss presented in [1]. This indicates that the intuition in [1]—that substantial reduction is achievable only at intermediate noise levels is not applicable to the linear noise scheduler,  raising questions about the applicability of log-normal sampling under this noise scheduler. This highlights the advantage of our method as we design sampling strategy considering the gradient variance and remains general across different noise schedulers. 2. With our VR-sampling strategy, we sample intermediate timesteps more frequently. As a result, in this range, the loss intervals are consistently smaller compared to the baseline method.

---

> ### Author Response · Authors · 2024-11-25
> **Reply to Reviewer Ud3A**
>
> > W3. performance near the convergence point.
>
> To further address your concern, we present results near the convergence points on the LSUN Bedroom-256 dataset using a linear noise scheduler and the SiT L/2 network. While the baseline achieves an FID of 4.09 at approximately 300k iterations, our VR-sampling method reaches an FID of 4.03 much earlier, at around 150k iterations. Furthermore, with VR-sampling, the FID improves to 3.91 at 250k iterations. We will try to add the results on ImageNet datasets in the final versions.
>
> | Iterations | Baseline   | VR-sampling |
> |------------|--------|--------|
> | 50k        | 11.94  | 10.05  |
> | 100k       | 4.91   | 4.50   |
> | 150k       | 4.35   | 4.03   |
> | 200k       | 4.23   | 4.01   |
> | 250k       | 4.10   | 3.91   |
> | 300k       | 4.09   | 3.98   |
> | 350k       | 4.16   | 4.04   |

---

> > ### Author Response · Authors · 2024-11-25
> > **Reply to Reviewer Ud3A**
> >
> > I truly appreciate the time and effort you’ve already dedicated to reviewing the work. Please let me know if there’s anything I can clarify.

---

> > ### Comment · Reviewer_Ud3A · 2024-11-25
> > **Further response**
> >
> > Thanks for the detailed response.
> >
> > W1 & W2. Could you please explain more clearly aobut the two phases in Theorem3.3?
> >
> > W3. My concern has been somewhat addressed. The authors should add the final experimental results in the next version.
> >
> > W4. My concern has been addressed.
> >
> > Q1 & Q2. My questions have been addreseed. Thanks for the detailed explaination.

---

> ### Author Response · Authors · 2024-11-25
> **Reply to Reviewer Ud3A**
>
> Thanks for your feedback. To address the concern about the connection between variance reduction (VR) and the observed acceleration in training, we provide a more detailed analysis as follows:
>
> **1. The Composition of Training Loss and Its Dynamics**
>
> In expectation, the training loss consists of two terms:
> \begin{align}
> \mathbb{E}[Training Loss] = \underbrace{L_\text{decay}}_{\text{Exponential Decay Term}} + \underbrace{\sigma^2 _{\text{noise}}} _{\text{Variance Term}},
> \end{align}
> where:
> - $L_\text{decay}$: an exponentially decaying term that dominates in the early phase of training. This term decreases rapidly as the model moves toward a local neighborhood of the global minimizer.
> - $\sigma^2_{\text{noise}}$: the variance term determined by the noise inherent in the sampling policy. This term dominates in later training stages when $L_\text{decay}$ becomes negligible.
>
> At the beginning of training, $L_\text{decay}$ dominates, leading to rapid reduction in the overall training loss. However, as training progresses and $L_\text{decay}$ diminishes, the variance term $\sigma^2_{\text{noise}}$ begins to dominate. At this point, the training loss stabilizes and appears not to decrease further. Importantly, this plateau does not mean that training has completely stalled; instead, the model's quality (e.g., image generation fidelity) continues to improve, albeit at a much slower pace. This slower improvement occurs because the gradients used to update the model become increasingly noisy, resulting in fewer effective updates.
>
> **2. The Benefits of VR-Sampling**
>
> The VR-Sampling strategy reduces the scale of $\sigma^2_{\text{noise}}$, which directly affects the dynamics of the two training phases:
>
> - **Extended Dominance of the Exponential Decay Phase**:
>   By reducing the variance of the noise term, VR-Sampling ensures that $L_\text{decay}$ remains the dominant component of the training loss for a longer period. This extends the phase of rapid loss reduction, allowing the model to approach the local neighborhood of the global minimizer more quickly.
>
> - **Reduced Gradient Noise in the Plateau Phase**:
>   When the variance term $\sigma^2_{\text{noise}}$ dominates, a smaller noise scale translates to less noisy gradients. This allows for more effective updates to the model parameters even in the later stages of training, where improvements to model quality are typically slow.
>
> The two advantages above combine to enable VR-Sampling to achieve higher-quality solutions more efficiently under the same computational budget.
>
> For W3, we will add the results on ImageNet near the convergence points in the final versions.

---

> > ### Comment · Reviewer_Ud3A · 2024-11-26
> > **Further discussion**
> >
> > Thanks for the authors' further response. Still on W1, I would think reducing $\sigma_{noise}^2$ leads to a better convergence point and $L_{decay}$ is not affected by VR-sampling which is still an exponentially decaying term. Thus, this theorems is showing a better convergence point rather than a better convergence speed? I still think the experimental results near convergence points are rather important.

---

> > > ### Author Response · Authors · 2024-12-01
> > > **Reply to Reviewer Ud3A**
> > >
> > > Thanks for your feedback.
> > >
> > > - Reducing $\sigma^2_\text{noise}$ directly improves the convergence point because the noise term limits the model’s ability to approach the true minimum. By minimizing this term, VR-sampling allows the model to converge closer to the ideal solution, resulting in lower final FID values at the same iterations.
> > > - Besides, VR-sampling enables faster reduction of the training loss by mitigating noisy gradient updates. This leads to reaching the same FID with fewer iterations, as evidenced by our experimental results in Fig.3-5. Thus, VR-sampling enhances the efficiency of the optimization process.
> > >
> > > Thanks. The results near the convergence points are indeed important but currently we can only present the results on smaller LSUN bedroom-256 datasets to show that the sampling strategy also works near the convergence points.

---

### Official Review · Reviewer_CPyS · 2024-11-01

**Soundness:** 3
**Presentation:** 2
**Contribution:** 3
**Rating:** 8
**Confidence:** 3

**Summary:**

1. This paper first analyzes the reason for training issues in Flow models and finds the important role of high-variance regions.

2. Based on the theoretical analysis, this paper proposes a variance reduction sampling strategy to sample more timesteps with high variance to accelerate model convergence.

3. The experimental results show that the acceleration is significant.

**Strengths:**

1. This paper provides the theoretical analysis for the training of the flow model.

2. Based on their findings and theoretical analysis, they propose a simple but quite effective method to accelerate the training.

3. I think this method can also be applied to other models easily and the conclusion will still be valid.

**Weaknesses:**

1. The complexity and overhead of Monte Carlo simulations. This work heavily depends on Monte Carlo simulation, which is complex and time-consuming. Although in the paper, the time is much faster than full training, it may still cause a large overhead. I think this paper should involve more analysis of several aspects that influence the speed and performance of simulation such as the higher number of samples and higher data dimension.

**Questions:**

1. Based the discussion and method in the paper, during training, we will sample more high-variance regions for training. Will this cause overfitting on high-variance regions?

---

> ### Author Response · Authors · 2024-11-22
> **Repy to Reviewer CPyS**
>
> >W1. The complexity and overhead of Monte Carlo simulations. This work heavily depends on Monte Carlo simulation, which is complex and time-consuming. Although in the paper, the time is much faster than full training, it may still cause a large overhead.  I think this paper should involve more analysis of several aspects that influence the speed and performance of simulation such as the higher number of samples and higher data dimension.
>
> **Reply:** We want to emphasize that Monte Carlo simulations are merely a preprocessing step and are not required for each iteration.  Furthermore, we provide recommended hyperparameter settings for logit-normal sampling in various scenarios, which ensure comparable acceleration performance, as shown in Table 3. These recommendations simplify further use under different settings.
>
> For a more detailed analysis of the speed and performance of Monte Carlo simulations, we have provided in Appendix A. Specifically,
>
> 1. **Impact of the Number of Data Samples and Gaussian Samples:** The time consumption is primarily influenced by the number of data samples $M$ and the number of Gaussian samples $K$. We list the time comsumption with different $M$ and $K$ in Table 4. As expected, increasing $M$ and $K$ leads to higher time consumption and the simulated curves will be more accurate. However, we find that the simulated curves under $M=200$ and $K=500$ works effectively in identifying critical timesteps for training, balancing efficiency and performance.
>
> 2. **Impact of Data Dimensions:** For the influence of data dimensions, we find that when we choose $M=200$ and $K=500$, the per timestep consumptions for ImageNet-256 (latent spaces of 4x32x32) and ImageNet-512 (latent spaces of 4x64x64) are 18.62s and 18.63s, respectively. This negligible increase in time consumption with higher data dimensions is likely due to the GPU implementation of our Monte Carlo simulations. Matrix computations in our simulations are not intensive for GPUs and increase the data dimension has negligible influence the computation time.
>
>
> Including Monte Carlo simulations in the overall training time results in only a small overhead.  For example, when we train SiT XL/2 on ImageNet-256 using 8×A100 GPUs, we take **6.9 days** for 950K iterations, while the Monte Carlo simulations take only **0.68 hours** on the same hardwares.  As the training iterations increases, the relative time cost of simulations becomes even smaller.
>
> In conclusion, Monte Carlo simulations introduce minimal overhead compared to the total training time and serve as an efficient preprocessing step to improve training performance. Their inclusion is valuable, especially when considering the acceleration benefits highlighted in our work.
>
> >Q1. Based the discussion and method in the paper, during training, we will sample more high-variance regions for training. Will this cause overfitting on high-variance regions?
>
> **Reply:**   As shown in Theorem 3.3, during training, the loss decreases rapidly until the model converges to a neighborhood of the global minimizer. The size of the neighborhood is determined by the peak variance scale of the sampled regions under the sampling policy. Our method reduces this peak variance scale, ensuring faster convergence to high-quality solutions.
>
> Naively sampling only high-variance regions could indeed lead to overfitting, as it would retain a global large peak variance scale. However, our sampling strategy maintains a balance by considering both high-variance and other regions (as our sampling distribution is ranging from 0 to 1), ensuring stable training and effective generalization.

---

> > ### Comment · Reviewer_CPyS · 2024-12-02
> > **Response to Rebuttal**
> >
> > Dear Authors,
> >
> > Thank you for your clarification. My concerns are mostly addressed. Therefore, I would like to increase my score.

---

### Official Review · Reviewer_22rT · 2024-11-01

**Soundness:** 1
**Presentation:** 2
**Contribution:** 2
**Rating:** 3
**Confidence:** 4

**Summary:**

The authors theoretically identify the high variance in loss gradient estimates at intermediate training timesteps in flow-based generative models, and this variance affects influences the convergence of the optimization process during training. To address this issue, they construct an upper bound for the average total gradient variance using a function related to the signal-to-noise ratio (SNR) and propose a Variance-Reduction Sampling (VR-sampling) strategy. This strategy prioritizes sampling timesteps from high-variance regions more frequently to improve training efficiency.

**Strengths:**

1. The authors propose an upper bound for the variance of loss gradient estimates during training of flow-based generative models.
2. With the VR sampling strategy, the number of training iterations required to achieve similar performance is significantly reduced compared to the baseline.

**Weaknesses:**

1. There is no comparison with the state-of-the-art methods, and the results in Table 2 are significantly worse than the current state-of-the-art performance.
2. The paper lacks a detailed introduction to the baseline used for comparison in the results tables.
3. The paper does not provide an analysis of how the VR reduction strategy enhances the qualitative results.
4. In Section 3.3, the sampling process in the proposed strategy is not clearly explained. The current description mentions calculating normalization, followed by the PDF and the inverse CDF, but it is unclear how the probability density function π(t) is derived. Is π(t) the result of the inverse CDF? Additionally, there is no information on the final distribution from which the samples are generated.

**Questions:**

What is the location parameter m in the caption of table 3?

---

> ### Author Response · Authors · 2024-11-22
> **Reply To Reviewer 22rT**
>
> >W1. The results in Table 2 are significantly worse than the current state-of-the-art performance.
>
> **Reply:** The claim that our results in Table 2 are significantly worse than the current SoTA methods is misunderstanding.  In our study, we adopt SiT [1] as our baseline, which represents the current SoTA methods _without any additional sampling or weighting strategies applied_.  Based on this baseline, we ran the officially released code of SiT with default hyperparameters and reported the FID and IS values at **400K** iterations.  For ImageNet 256 under a linear noise scheduler, we achieve a baseline FID-50K (reported as FID-10K in our paper) of **17.3** using SiT XL/2, which is comparable to the reported value of **17.2** at 400K iterations in [1] (see Table 1 in [1]).  Furthermore, when using the officially released SiT code with our proposed VR-sampling strategy, we achieve an FID-50K of **13.5** at 400K iterations—showing a significant improvement over the baseline.
>
> The primary reason we report results at 400K iterations is that our method is **generally applicable to any noise scheduler (including both diffusion and flow model settings) and data dimension**. To validate its effectiveness experimentally, we conducted at least 9 experiments across various settings. Running all these experiments to full convergence (around **7000K** iterations in ImageNet-256) would be highly resource-intensive. For instance, a single experiment on ImageNet256 at convergence would take over one month using 8×A100 GPUs, which is infeasible given our limited computational resources. Therefore, we report FID values at 400K iterations for ImageNet-256 and 512, a practical and consistent choice that aligns with the settings used in comparable works, such as [2,3]. This approach ensures a fair and resource-efficient evaluation of our method.
>
> To further validate the effectiveness of our method, we have included additional experimental results in the revised paper showing FID values at approximately **950K** iterations (which required 6.5 days) for ImageNet-256 under the flow model setting. Refer to the newly added FID curves in Fig.19 in the Appendix for more details. From these results, we observe that the SiT baseline achieves an FID value of 16.0 at 950K iterations, while our method achieves an FID of 16.1 at only 600K iterations, demonstrating the efficiency of our approach. We will try to include additional results closer to 7000K iterations in the final version of the paper.
>
> >W2. There is no comparison with the state-of-the-art methods.
>
> **Reply:** Regarding comparisons with _the SoTA performance involving sampling or weighting strategies_, previous works primarily focus on the diffusion model setting (see Sec.5 Related works).
> - In our paper, under the diffusion model setting, we compare our VR-sampling strategy with the current SoTA method--SpeeD [2] as shown in **Figure 7**.
> - For flow-based models, we compare against the logit-normal sampling method (with default hyperparameters) introduced in Stable Diffusion 3, as illustrated in **Figure 6**.
>
> Our method performs better under both two model settings.
>
> [1] SiT: Exploring Flow and Diffusion-based Generative Models with Scalable Interpolant Transformers. 2024
>
> [2] A closer look at time steps is worthy of triple speed-up for diffusion model training. 2024.
>
> [3] Towards faster training of diffusion models: An inspiration of a consistency phenomenon. 2024
>
> >W3. The paper lacks a detailed introduction to the baseline used for comparison in the results tables.
>
> **Reply:** As mentioned in our reply to W1-2, we selected SiT [1] as our baseline, which unifies diffusion models and flow models under the interpolant framework. We illustrate the details of SiT in Sec.2. We also compared with two SoTA methods (SpeeD [2] and logit-normal sampling) in Fig.7 and Fig.6, respectively. All relevant experimental configurations are detailed in Table 6 for clarity.
>
> >W4. The paper does not provide an analysis of how the VR reduction strategy enhances the qualitative results.
>
> **Reply:** Due to space constraints, we presented our qualitative results in **Figures 13-15** in Appendix B.4. Additionally, we have added more qualitative results with higher resolution in **Figures 16-17** for better visualization. From these results, we observe that our VR-sampling strategy significantly improves image quality, especially during earlier iterations. For example, in the red panda images shown in Figure 16, at 200K iterations, our method produces normal-looking images with no distorted faces, whereas the baseline still exhibits noticeable distortions even at 400K iterations.

---

> > ### Comment · Reviewer_22rT · 2024-11-25
> >
> > W1: In the caption of Table 2, the results are reported at 400k iterations on the ImageNet 256 dataset. The best result (17.15) is from the cosine scheduler. However, no significant improvement is observed compared to the 17.2 reported at 400k iterations as well in Table 1 of the SiT paper. I consider the Table 2 shows the main results, but the corresponding text is only one sentence in line 382. There is still no clear information on how the baseline results in Table 2 were obtained. This lack of detailed discussion raises concerns about the contribution of speedup and the reliability of all the curve results presented in the figures.
> >
> > The state-of-the-art performance in SiT is 2.06, and many methods listed in Table 7 of SiT achieve results under 5. However, this paper provides no comparison with those methods. Therefore, the authors cannot guarantee that the final results at 7000k will surpass the SiT benchmark of 2.06. Although the authors said they would include these results later because of the long training time, I can only evaluate the information currently presented in the paper and cannot assume that the unavailable results will be helpful..

---

> ### Author Response · Authors · 2024-11-22
> **Reply To Reviewer 22rT**
>
> >W5. In Section 3.3, the sampling process in the proposed strategy is not clearly explained. The current description mentions calculating normalization, followed by the PDF and the inverse CDF, but it is unclear how the probability density function π(t) is derived. Is π(t) the result of the inverse CDF? Additionally, there is no information on the final distribution from which the samples are generated.
>
> **Reply:** Thank you for pointing this out. We have clarified the sampling process in the revised paper as follows:
>
> After using Monte Carlo simulations to obtain variance curves ranging from 0 to 1, these curves are typically non-normalized and cannot directly serve as valid probability density functions (PDFs). To address this,
>
> (1) We first normalize the simulated curve by calculating the total area under the curve (via integration) and dividing the curve by the total area.
>
> (2) We compute the cumulative sum of the normalized curve to derive the cumulative distribution function (CDF) $F(t)$.
>
> (3) Using the CDF, we construct an interpolation function to approximate its inverse $F^{-1}(u), u\sim Uniform(0,1)$.
>
> (4) The inverse CDF maps a uniformly distributed random variable to the domain of $t$, ensuring that the resulting samples follow the distribution defined by the probability density function (PDF) $\pi(t)$, where $\pi(t) = F'(t)$.
>
> Here the inverse CDF serves as a crucial tool for sampling from the desired density. Our final simulated sampling density is $\pi(t)$, and we provide visualizations of its simulated distribution in Fig. 1 (d)-(f) for better understanding.
>
> **Question:**
> >Q1. What is the location parameter m in the caption of table 3?
>
> **Reply:** We introduce this definition in the definition of logit-normal sampling $logit-normal(m, s)$, see equation in line 427. $m$ and $s$ are the mean and standard deviation of the variable's natural logarithm, not the expectation and standard deviation of variable.

---

> ### Author Response · Authors · 2024-11-25
> **Reply To Reviewer 22rT**
>
> We truly appreciate the time and effort you’ve already dedicated to reviewing the work. Please let us know if there’s anything we can clarify.

---

> ### Author Response · Authors · 2024-12-01
> **Reply To Reviewer 22rT**
>
> > 1. Clarification on Table 2 Results
>
> Thank you for your feedback. We would like to clarify how the results in Table 2 were obtained. The performance of SiT is influenced by both **training** and **inference** settings. In this work, we focus on training acceleration, while just keeping the inference methods consistent across different training experiments.
>
> - Training Settings: The training configurations were detailed in Appendix A.2, and the evaluation (inference) settings were described in Section 4.1 (Evaluation part).
> - Inference Settings: Across all training on ImageNet datasets, we use **250 sampling steps** with the **ODE-dopri5 solver** to generate **10K images** for evaluation. In comparison, SiT reports its SoTA result using **250 sampling steps** with the **SDE-Euler-$w^{KL}$-Mean-0.04** solver to generate **50K images**.
>
> To ensure reliability, we re-implemented the SiT baseline under the same inference methods. Using the **linear noise scheduler**, we reproduced the reported baseline FID of **17.2** at 400K iterations, confirming the correctness of our implementation. In SiT paper, they claim that the best results for SiT XL/2 are from velocity prediction under the linear interpolants under SDE sampler (see the line above Sec. 3.3 in SiT), which is the same setting we use now. Applying our VR-sampling method in SiT and inference under this same settings improved the FID to **13.5**, which much better than the SoTA value in SiT under 400K iterations and demonstrate the effectiveness of our method in accelerating training. Besides, we report the FID values of SiT under **cosine noise scheduler** using **SDE-250-Euler-decreasing-Mean-0.04** and obtain the baseline **17.85**. Under the same setting, using our VR-sampling we obtain **13.64**.
>
> To show the inference methods doesn't influence on the training acceleration, we originally compared the results of ODE-dopri5 of 250 sampling steps and 30 sampling steps in Table 7 in Appendix, which show the same level of training acceleration improvement over FID. We appreciate the reviewer for highlighting this point. While reporting SoTA values is not the primary focus of our work, we acknowledge the importance of providing these results for completeness.
>
>
> >2. Response to Table 7 [1] Comparison
>
> We want to emphasize that our work aims to **accelerate training** by reducing computation while achieving comparable or better performance at earlier training stages. Our method improves the sampling strategy in existing SoTA models, such as SiT, rather than introducing a new network architecture or training method. For example, we also applied VR-sampling to DiT, demonstrating acceleration performance that surpasses the current SoTA method (SpeeD) in DiT at 400K iterations, as shown in Fig. 7. The key intuition behind our work is to identify the critical timesteps during the training of generative models and allocate more emphasis to these sampling steps. Current large models, such as SD3 and LuminaT2X [1], are also exploring the use of importance sampling ideas to accelerate training. Our work provides a systematic study and deeper understanding of this approach.
>
>
> Although we will later provide results near convergence points for ImageNet, we emphasize that our method on training acceleration has broad practical applicability. In many practical scenarios, it is infeasible to train every model or configuration to full convergence due to the high computational cost. At the same time, evaluating models  in the early training phase can lead to unreliable performance assessments. Our approach addresses this gap by allowing models to rapidly approximate their performance, making it possible to identify promising candidates early.
>
> [1] Lumina-T2X: Transforming Text into Any Modality, Resolution, and Duration via Flow-based Large Diffusion Transformers

---

### Note · Authors · 2025-02-16

I have read and agree with the venue's withdrawal policy on behalf of myself and my co-authors.

---

### Meta-Review · Area_Chair_M5zk · 2024-12-19

**Metareview:**

This paper theoretically identifies the higher variance in the loss gradient estimates as the root cause of difficulty in learning at intermediate timesteps. To help solve this, the author provides a VR-sampling strategy that samples the timesteps in high-variance regions more frequently.

This paper provides some theoretical analysis throughout the paper. However, the comparison with the state-of-the-art methods is relatively weak. There are other concerns about the convergence of this VR-sampling approach.

The reviews are mixed (3, 5, 8, 8).

**Additional Comments On Reviewer Discussion:**

One of the negative reviewers is still concerned about the explanation and analysis for how VR-sampling changes/improves the convergence. Another negative reviewer mentioned that the comparison with the state-of-the-art methods is weak. These two concerns seems important and still haven't been fully addressed after rebuttal.

---

### Decision · Program_Chairs · 2025-01-22

Reject